# A direct interareal feedback-to-feedforward circuit in primate visual cortex

Caitlin Siu[1], Justin Balsor[1], Sam Merlin [1,2], Frederick Federer[1] & Alessandra Angelucci [1✉]

The mammalian sensory neocortex consists of hierarchically organized areas reciprocally connected via feedforward (FF) and feedback (FB) circuits. Several theories of hierarchical computation ascribe the bulk of the computational work of the cortex to looped FF-FB circuits between pairs of cortical areas. However, whether such corticocortical loops exist remains unclear. In higher mammals, individual FF-projection neurons send afferents almost exclusively to a single higher-level area. However, it is unclear whether FB-projection neurons show similar area-specificity, and whether they influence FF-projection neurons directly or indirectly. Using viral-mediated monosynaptic circuit tracing in macaque primary visual cortex (V1), we show that V1 neurons sending FF projections to area V2 receive monosynaptic FB inputs from V2, but not other V1-projecting areas. We also find monosynaptic FB-to-FB neuron contacts as a second motif of FB connectivity. Our results support the existence of FF-FB loops in primate cortex, and suggest that FB can rapidly and selectively influence the activity of incoming FF signals.

[1] Department of Ophthalmology and Visual Science, Moran Eye Institute, University of Utah, Salt Lake City, UT, USA. [2] Present address: Medical Science, School of Science, Western Sydney University, Campbelltown, NSW, Australia. ✉email: alessandra.angelucci@hsc.utah.edu

In the neocortex, sensory information is processed within hierarchically organized areas reciprocally connected via feedforward (FF) and feedback (FB) circuits[1,2]. FF connections carry information from lower to higher-level areas. As information ascends through the cortical hierarchy, neuronal receptive fields (RFs) become tuned to increasingly complex stimulus features, and an increasingly abstract representation of sensory inputs is achieved. FF connections are reciprocated by FB connections sending information from higher to lower areas. This hierarchy is further organized into parallel processing streams, so that cortical areas within each stream are functionally specialized to process specific attributes of a sensory stimulus[3,4]. Reciprocal FF–FB connections between pairs of cortical areas are found throughout the neocortex of all mammalian species, suggesting they carry out a fundamental computation, but their role remains poorly understood.

Traditional FF models of sensory processing postulate that FF connections mediate the complexification of RFs, and that object recognition occurs largely independently of FB signals[5–7], the latter purely serving attentional selection. In contrast, several theories of hierarchical computation postulate that most of the computational work of the cortex is carried out by information going back and forth over looped FF–FB circuits between pairs of interconnected areas[8–16]. The exact computation performed by these loops depends on the specific theory, but many of these theories require FF–FB loops to occur between neurons in different areas processing similar stimulus attributes, albeit at different levels of abstraction. Whether this anatomical organization of FF–FB loops indeed exists in the cortex remains unclear. It is well established that most cortical areas possess reciprocal FF and FB connections[17,18]. However, since each area projects too, and receives inputs from, multiple areas, it is less clear whether FF and FB connections selectively contact the neurons that are the source of their reciprocal areal input, or rather unselectively contact different projection neurons in their target area. It is also unclear whether these cortico-cortical loops occur via direct monosynaptic contacts between FF and FB projection neurons, or indirectly via local excitatory or inhibitory neurons, or both. Recent studies have shown that in mouse primary visual cortex (V1), only a fraction of cortical projection neurons forms area-specific monosynaptic FF–FB loops[19] and that these loops may only occur between deep layer neurons[20]. It remains unknown whether similar rules of cortico-cortical connectivity apply to higher mammals.

In cats and primates, inter-areal FF projections are highly area-specific, much more so than in rodents. For example, only a small percent of FF-projecting V1 neurons send a common input to multiple extrastriate areas via bifurcating axons[21–23]. However, it is less clear whether FB projections show similar area specificity, and whether they influence FF projection neurons directly or indirectly. On the one hand, previous reports that neurons in the extrastriate cortex sending FB projections to V1 and the secondary visual area (V2) contain substantial amounts of axonal bifurcations[21,24–26] and form diffuse terminations[27–29] suggest that FB may not selectively contact the neurons that are the source of their areal input. On the other hand, recent demonstrations of clustered and specific FB terminations in primate V1[30] (see also ref. [31]) suggest the opposite.

To address this question, here we adapted viral-mediated monosynaptic input tracing methods[32] to label the inputs to V1 neurons sending FF projections to V2 in the macaque visual cortex. If FB connections did not selectively contact the neurons that are the source of their areal FF input, one would expect to find inputs to V1 neurons projecting to V2 (V1→V2) to arise from multiple extrastriate areas known to project to V1. If, on the other hand, FB connections to V1 were area-specific, one would

expect to find FB inputs to V1→V2 cells to arise only from V2. Consistent with the latter scenario, here we find that V1→V2 neurons receive monosynaptic inputs from V2 FB neurons, but not from neurons in other extrastriate areas known to also send FB projections to V1. We also find evidence for direct cortico-cortical FB-to-FB contacts. These results suggest that FB can rapidly and selectively influence the activity of incoming FF signals, and support the existence of area-specific FF–FB loops in the primate early visual cortex.

## Results

**Monosynaptic input tracing in macaque cortex.** We adapted viral-mediated monosynaptic input tracing or TRacing Inputs and Outputs (TRIO[32,33]) to identify, in the macaque visual cortex, direct presynaptic inputs to V1→V2 neurons. We used an intersectional viral strategy based on three different viral vectors injected at different times (Fig. 1a; see Methods). Specifically, we injected in V1 a mixture of two Cre-dependent adeno-associated viruses serotype 9 (AAV9), one carrying the gene for the avian tumor virus receptor A (TVA receptor for EnvA) fused with the red fluorescent protein mCherry (AAV9-CAG-FLEX-TVAmCherry), the other carrying

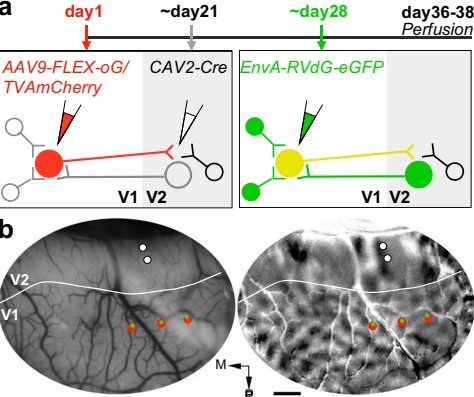

**Fig. 1 Monosynaptic input tracing in macaque visual cortex: experimental design. a** Viral injection timeline and experimental design. Left: V1→V2 neurons express mCherry (red cells), TVA, and oG, after double infection with AAV9-vectors (injected in V1) and CAV2-Cre (injected in V2). Right: After additional infection with EnvA-RV*dG*-eGFP (injected at the same V1 sites as AAV9), V1→V2 neurons previously infected with AAV9 additionally express eGFP, becoming double-labeled (yellow starter cells). After *trans*-synaptic RV*dG*-eGFP infection, V1 and V2 cells presynaptic to the V1 starter cells express eGFP (green cells). Cells that are not co-infected with both CAV2 and AAV9 remain unlabeled. **b** Injection plan. In vivo OI of V1 and V2 in one example case (MK405). Left: Image of the cortical surface vasculature encompassing V1 and V2. Solid white contour: V1-V2 border based on the orientation map. The surface vasculature is used as a reference to target viral injections to matching retinotopic positions in V1 and V2. To ensure retinotopic overlap of the V1 and V2 injections, multiple AAV (up to 3) injections (red dots), spaced about 1 mm mediolaterally, are made in V1, and up to 2 CAV2 injections (white dots), spaced about 300 μm anteroposteriorly, are made in V2. RV*dG* injections (green dots) are targeted to the same locations as the AAV injections, using as guidance images of the surface vasculature taken at the time of the AAV injections. Right: Orientation difference map of V1 and V2 obtained by subtracting responses to achromatic luminance gratings of two orthogonal orientations. Orientation and other functional maps are used to identify the V1/V2 border, so as to target injections to the appropriate areas. For example, in the orientation map, V2 can be distinguished from V1 due to its "stripy" pattern and larger orientation domains. M: Medial; P: posterior. Scale bar: 1 mm. Optical maps in (**b**) are representative of three independent cases.

the gene for the optimized rabies virus glycoprotein (oG[34]) (AAV9-CAG-FLEX-oG-WPRE). After about 3 weeks, necessary for the AAV genome to concatermerize (i.e., generate multiple bound copies of its genome), a canine adenovirus type 2 carrying Cre-recombinase (CAV2-CMV-Cre[33,35]) was injected in V2 at retinotopic locations matched to those of the V1 injections. CAV2 is a retrograde vector that rapidly transcribes Cre-recombinase in local V2 neurons and V1→V2 neurons projecting to the CAV2 injection site in V2, reaching maximum expression in 5–7 days. In the presence of Cre, in V1 only the V1→V2 cells previously infected with AAV9-FLEX vectors express mCherry (thus, turning red; Fig. 1a), TVA, and oG. About one week after the CAV2 injections, EnvA-pseudotyped, G-deleted rabies virus (RVdG) carrying the gene for green fluorescent protein (eGFP) (EnvA-RVdG-eGFP) was injected in V1 at the same location as the AAV9 injections. Since the EnvA ligand binds exclusively to the TVA receptor, which is not otherwise native in the primate brain, RVdG can only infect cells that express TVA. This results in the expression of GFP in TVA-expressing V1→V2 cells, which become double-labeled in red and green (yellow "starter" cells in Fig. 1a). Moreover, the presence of oG in the starter cells, allows RVdG complementation and retrograde monosynaptic spread of the rabies virus, with consequent GFP expression in the presynaptic input cells, which turn green (Fig. 1a). As the input cells do not express oG, RVdG cannot further spread trans-synaptically beyond these neurons. To identify the V1/V2 border and ensure retinotopic overlap of the injections in V2 and V1, we used in vivo intrinsic signal optical imaging (OI) as guidance and made 2–3 injections of the AAV9 and RVdG vectors in V1, and 1–2 injections of CAV2-Cre in V2 as schematically shown in Fig. 1b (see Methods). The V1 injections spanned all cortical layers, while the V2 injections were centered in layer (L)4, where the bulk of V1 FF projections terminate. Injection sites in V1 and V2 for an example case (MK405) that received 3 AAV and RVdG injections in V1 and 2 CAV2 injections in V2 (Supplementary Table 1) are shown in Figs. 2 and 3.

Figure 2a shows low-power images of the V1 injection sites in a tangential section through V1 L2/3 stained for cytochrome oxidase (CO) after being imaged under fluorescent illumination. In CO staining, pipette tracks can often be identified as discolorations visible across multiple sections. In the CO-stained section (Fig. 2a top panel), five distinct small pipette tracks are visible (black arrows), of which, the top three correspond, under fluorescence illumination, to RVdG-GFP injections, and the bottom two to AAV injections (Fig. 2a bottom panel). A third separate AAV injection is not discernible in CO, likely because it overlapped with the RVdG injection. As mCherry expression in V1 can only occur in cells that co-express Cre-recombinase, the larger cluster of mCherry-labeled cells (red) is visible at and around the middle AAV injection site indicates this injection was in good retinotopic overlap with the CAV2-Cre injection sites in V2. Instead, the sparse red label nearby the medial and lateral AAV injections indicates these injections were not well matched retinotopically to the V2 injection sites. In contrast to mCherry, some GFP expression can occur independently of Cre (due to small amounts of local TVA "leak"), but only locally at the injection site (see Results below for a more extensive discussion of TVA leak). This explains why all three RVdG injections are visible, even if the medial and lateral injections were not in good retinotopic overlap with the V2 injection sites. Double-labeled (yellow) cells are only visible nearby the middle injections, where the RVdG and AAV injection sites overlap (Fig. 2b, c). As the V1 injection sites encompassed all layers, in addition to L2/3, double-labeled cells were also found in all other layers known to project to V2, namely 4A–B, 5 (Fig. 2d, e), and 6 (Fig. 2d, f).

Figure 3 shows the V2 injection site for the same example case. The injection site (Fig. 3a, b black arrow) is recognizable as a region of small damage and CO discoloration along the pipette track, as well as by the presence of mCherry-labeled axon terminals of V1→V2 neurons in L4 and lower 3 within about 1 mm of the V2 pipette track (Fig. 3 and Supplementary Figs. 1 and 3b). The mCherry fiber label in V2 also confirmed the retinotopic overlap of the V2 and V1 injections. In the example case, the above injection protocol resulted in GFP-labeled input cells within V1 (Figs. 2 and 7a–c) and in V2 (Fig. 3), with no GFP label observed in other cortical or subcortical structures. In V2, GFP-labeled cells were located in the layers known to send FB projections to V1, namely the superficial layers, where they appeared to be more numerous in L3, and the deep layers, where they appeared to be more numerous in L5 (Fig. 3 and Supplementary Fig. 1).

We next describe the quantitative analyses of the distribution of double-labeled cells in V1 and of GFP-labeled cells in V2 for all cases.

**V1→V2 neurons receive area-specific monosynaptic FB inputs.** To label monosynaptic inputs to V1→V2 neurons, we performed TRIO experiments using the protocol described above in three macaque monkeys (Supplementary Table 1).

Figure 4a shows the laminar distribution of double-labeled (yellow) cells in V1 for each case. Supporting the cell specificity of our viral approach, in all cases, yellow cells were located only in the layers known to project to V2, i.e., the superficial and deep layers, but not L4C, which does not project out of V1. In all cases, the vast majority of yellow cells resided in the superficial layers, consistent with the known laminar origin of V1-to-V2 projections which arise predominantly (96–98%) from the superficial layers[36,37]. There were small variations in the distribution of yellow cells across cases. In the case of MK405, all layers are known to send outputs to V2 contained yellow cells, but 90% of them resided in L2/3. In cases MK382 and MK379, instead, yellow cells were located in all V2-projecting layers except L6, and while also more abundant (~80%) in the superficial layers, a significant fraction (~20%) was located in L5. The majority of superficial layer yellow cells were located in L2/3 in cases MK379 and MK405, while they were more evenly distributed across superficial layers in cases MK382. Across the population, on average $62.2 \pm 16.5\%$ of V1→V2 yellow cells were located in L2/3, and 85% were located in the superficial layers (Fig. 4b).

We measured the spatial spread of the yellow cells in the tangential domain of V1 along an axis parallel to the V1/V2 border, pooled across layers (Fig. 4c; see Methods). The maximum cortical spread of yellow cells in V1 ranged between 1.78 and 3.75 mm (mean ± sem: $3 \pm 0.62$ mm) depending on the size of, and overlap between, the viral injection sites (see Supplementary Table 1). After removing the tails of the distributions (2.5% of cells at the extreme of each distribution), the spread ranged between 1.75 and 2.75 mm (mean ± s.e.m: $2.42 \pm 0.33$ mm).

In all three cases, we found GFP-labeled cells in V2 superficial and deep layers, i.e., the layers of origin of FB connections to V1. We quantified the distribution of these cells across V2 layers for each case (Fig. 5a). Because the border between V2 L2 and L3 is not easily identifiable in the tangential sectioning plane, we did not attempt to distinguish between these two layers. In two cases (MK405 and MK382), GFP-labeled cells were found in almost similar amounts in superficial and deep layers. Instead, in case MK379 the majority of GFP-labeled cells were in deep layers. In all cases, GFP-labeled cells in deep layers were located in both L5

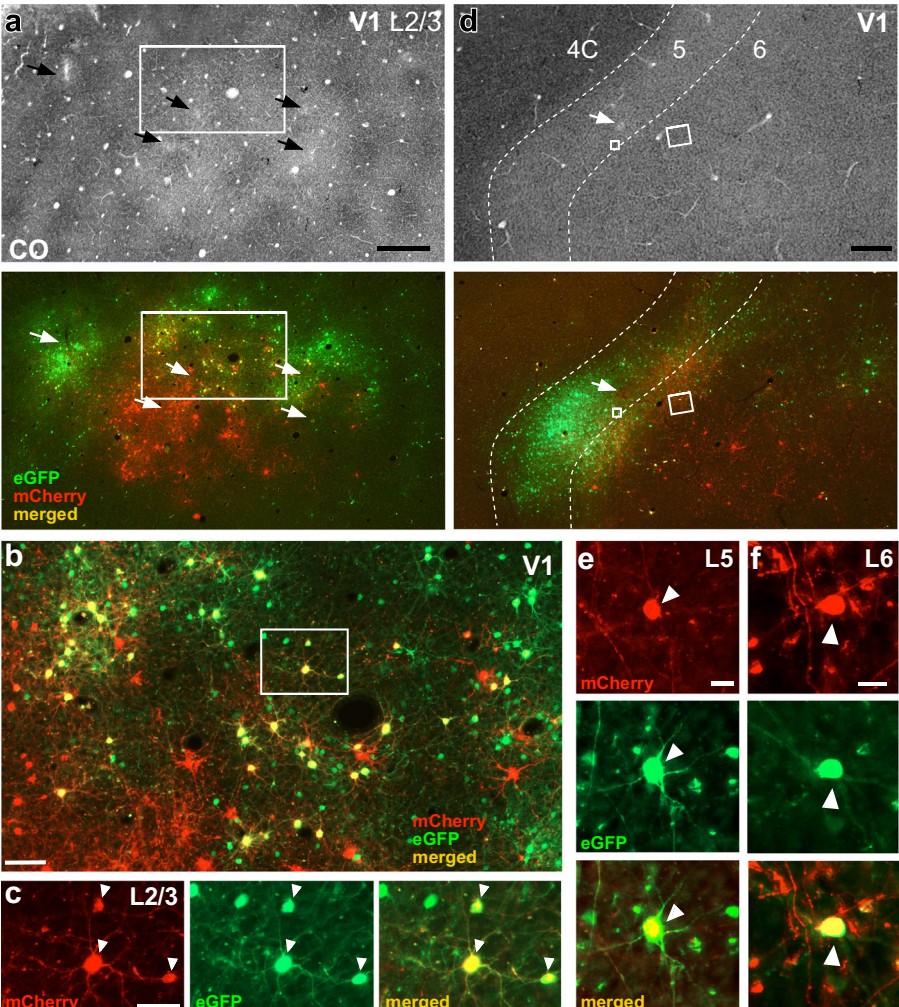

**Fig. 2 V1 injection sites. a** Case MK405. Image of a single tangential section through V1 L2/3 stained for CO (Top) after being imaged for mCherry and GFP fluorescence (Bottom). The merged channel shows double-labeled (yellow) "starter" V1→V2 cells. Arrows point to the V1 injection sites in both sections. The region inside the white box is shown at higher magnification in panel (**b**). Scale bar: 500 µm. **b** Higher magnification of the V1 region inside the box in panel (**a**). Red cells: V1→V2 neurons co-infected with CAV2 and AAV9-TVAmCherry, but not with RV$dG$. Yellow cells: starter V1→V2 cells double-labeled due to triple infection with CAV2-Cre, AAV9-TVAmCherry, and RV$dG$-GFP. Of these double-labeled cells only those that were additionally infected by AAV9-oG act as "starter" cells. Green cells: cells sending monosynaptic input to the starter V1→V2 cells (but some local V1 green label is due to TVA "leak"—see Results and Supplementary Fig. 2). Scale bar: 100 µm. Cells in the boxed region are shown at higher magnification in panel (**c**). **c** Higher magnification of 3 double-labeled V1→V2 cells (arrowheads) shown under mCherry (Left) or GFP fluorescence (Middle), and merged (Right). Scale bar: 50 µm. **d** Image of a single tangential section through V1 L4C-6 stained for CO (Top) and imaged for fluorescent signals (Bottom) in the same case as in (**a–c**). Yellow cells inside the small and large white boxes are shown at higher magnification in panels (**e**) and (**f**), respectively. Dashed contours delineate layer boundaries, and layers are indicated at the top. Scale bar: 500 µm. **e, f** V1→V2 starter cells (arrowheads) in L5 (**e**) and L6 (**f**), shown under mCherry (Top) or GFP (Middle) fluorescence, and merged (Bottom). Scale bars: 20 µm. Results in (**a–f**) are representative of injection sites made in three independent cases with similar results.

and 6 but were much more numerous in L5. The L5 origin of these FB inputs is further demonstrated in a series of tangential sections in Supplementary Fig. 1. In two cases, a few GFP labeled cells were also found in V2 L4; these amounted to only 3 cells in MK405 (0.5% of total V2 GFP label) and 20 cells in MK379 (1.7% of V2 GFP label). As L4 in primates, the early visual cortex is not a source of FB projections[29], these GFP-labeled cells in V1 L4 are unlikely to represent FB inputs to V1; their likely origin is discussed in a later section of the Results. Across the population, on average $54.3 \pm 10.2\%$ (s.e.m.) of GFP-labeled V2 cells were located in L5, $4.9 \pm 0.6\%$ in L6, and $40.1 \pm 10.1\%$ in L2/3 (Fig. 5b).

We measured the spatial spread of GFP-labeled cells across the tangential domain of V2, along an axis parallel to the V1/V2

border, pooled across layers (Fig. 5c; see Methods). The spatial spread of the V2 FB inputs was very extensive, ranging between 5.7 and 13.5 mm (mean max spread across the 3 cases $9.9 \pm 2.28$ mm). After removing the tails of the distribution, the range of spread was 4–7.8 mm (mean ± sem: $5.6 \pm 1.12$ mm). For each case, we calculated a ratio of the spatial spread of the V2 FB inputs to that of the V1→V2 starter cells. The latter ranged between 2.8 and 3.9 mm (mean ± s.e.m: $3.3 \pm 0.32$ mm) or, after removing the tails of the distributions, 1.8 mm and 2.8 mm (mean ± s.e.m: $2.3 \pm 0.29$ mm). Thus, the spread of the V2 FB inputs is about 2–3 times the size of the V1 region to which they project. These results are consistent with previous reports that V2 FB neurons convey information from a larger region of visual space to their target V1 cells[38].

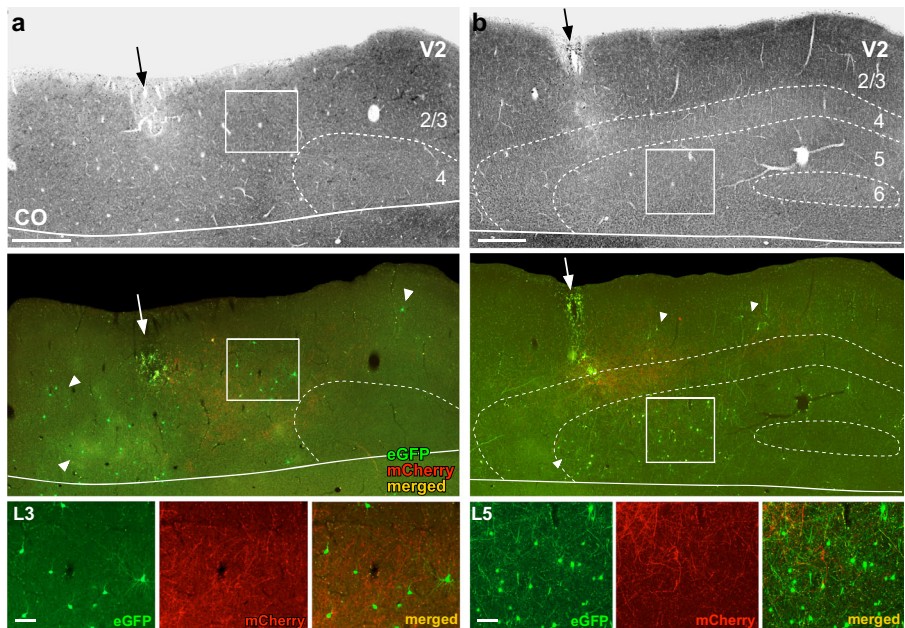

**Fig. 3 V2 injection sites and resulting GFP-label in V2. Case MK405. a** Image of a tangential section through V2 L1–4 stained for CO (Top) after being imaged for mCherry and GFP fluorescence (Middle). The middle panel shows the merged fluorescent channels. Arrows: V2 injection sites; white arrowheads point at some GFP-labeled input neurons. Red fibers are the terminals of V1→V2 neurons in L3–4. Solid white contour: V1/V2 border (V1 is below the border). The region inside the white box is shown at higher magnification in the bottom panel. Other conventions as in Fig. 2. Bottom: higher magnification of GFP-labeled V2 input cells in L3, shown under GFP or mCherry fluorescence, and merged, as indicated. Scale bar: 100 μm. **b** Same as in panel (**a**), but for a tangential section through V2 L1–6 showing denser GFP label in L5. Scale bars in (**a, b**): 500 μm (Top, Middle), 100 μm (Bottom). Supplementary Fig. 1 shows additional images illustrating the distribution of the GFP label across V2 layers. Results in (**a, b**) are representative of injection sites made in three independent cases with similar results.

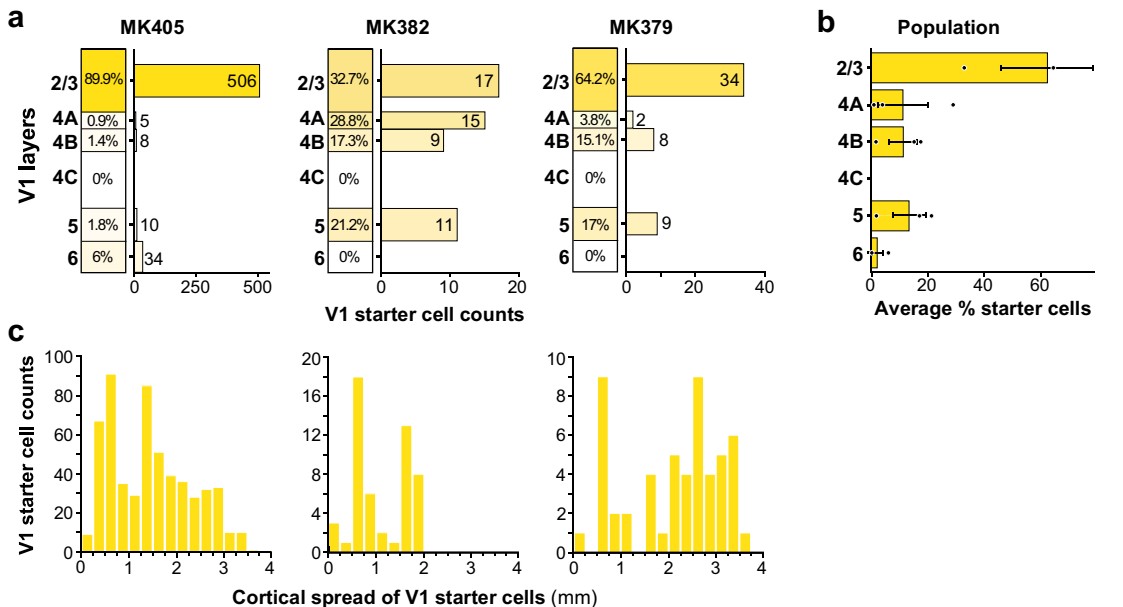

**Fig. 4 Laminar and tangential distribution of double-labeled "starter" cells in V1. a** For each of the three cases, we show the percentage (left column) and the number (right bar graph) of V1→V2 double-labeled cells across V1 layers. **b** Average percent of double-labeled cells across V1 layers for the population ($n = 668$ cells in 3 independent animals). Error bars: s.e.m. **c** Distribution of double-labeled V1 cells across the tangential domain of V1, collapsed across layers, for each case. Zero represents the location of the most medial double-labeled cell and the bin with the largest number represents the most lateral location of double-labeled cells.

The population average ratio of a total number of V2 GFP-labeled input neurons to a total number of V1→V2 starter cells (pooled across layers) was 8.7 ± 6.98; the average ratio of GFP-labeled V2 FB cells in each layer to the total number of V1→V2 starter cells pooled across V1 layers was 2.1 ± 1.2 for L2/3, 6.1 ± 5.4 for L5, and 0.3 ± 0.3 for L6 (Fig. 5d). The variability in this ratio across cases was due to case MK379, for which there were about 23 total V2 FB input cells per V1 starter cell, while this

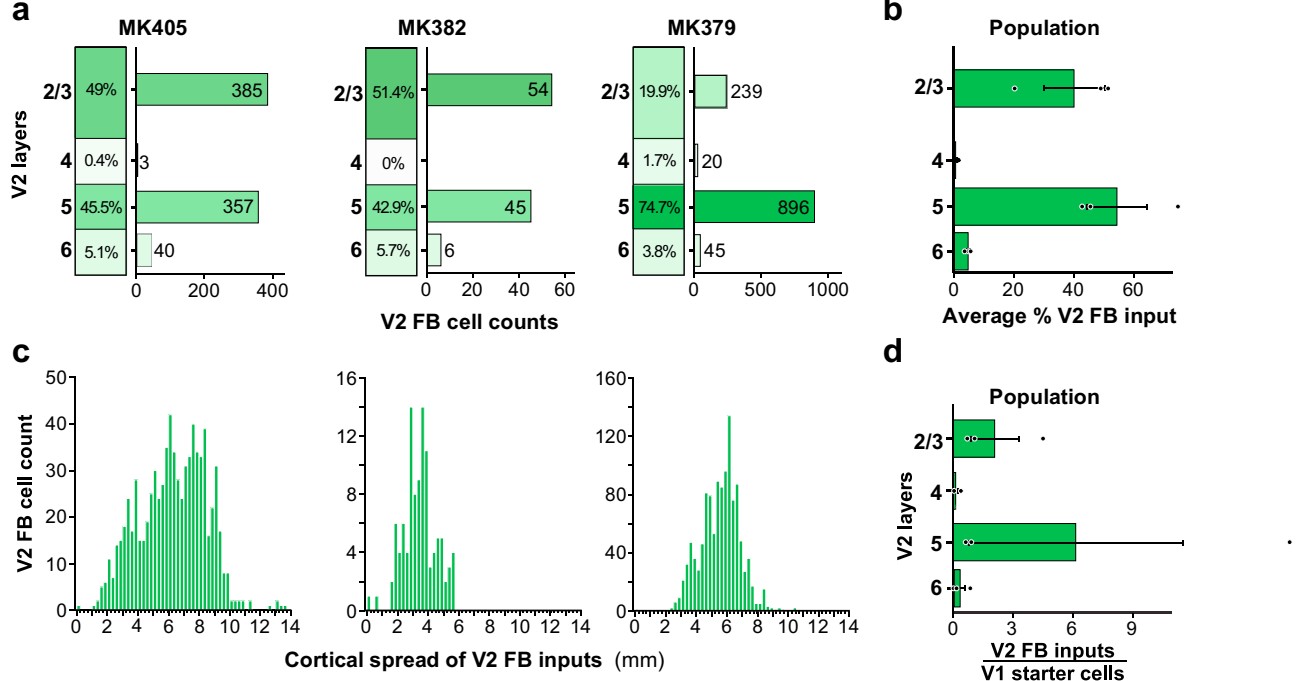

**Fig. 5 Laminar and tangential distribution of monosynaptic V2 FB inputs to V1→V2 cells. a** Percent and number of GFP-labeled cells across V2 layers for each of the three cases. **b** Population average percent ± s.e.m. of GFP-labeled cells across V2 layers (n = 2090 cells in three independent animals). **c** Distribution of GFP-labeled cells across the tangential domain of V2 pooled across layers. Other conventions as in Fig. 4. **d** Population average ratio of V2 input cells in each layer to the total number of V1→V2 starter cells (pooled across layers; n = 3 independent animals). Error bars: s.e.m.

ratio was about 2:1 for the other two cases. The possible source of this variability is discussed in the following section of the "Results".

As V1 receives FB connections not only from V2, but also from higher extrastriate cortical areas, including MT, V3, V4, and V6[18,29,39], to determine whether FB contacts with V1→V2 neurons are area-specific, we searched for fluorescent labels throughout the cortex anterior to V2, excluding only prefrontal cortex. In two cases (MK405 and MK382), we found no GFP-labeled cells in the cortex anterior to V2, while in the third case (MK379), we found a total of 7 GFP-labeled cells in the extrastriate cortex anterior to V2 (in areas V3, V3A, and MT), corresponding to 0.6% of the total number of GFP-labeled FB cells in cortex anterior to V1 (Fig. 6). These results indicate that monosynaptic FB contacts with V1→V2 neurons are highly area selective, and support the existence of highly specific FF–FB loops. In the next section of the Results, we present evidence supporting the interpretation that the few GFP-labeled FB neurons found outside of V2 in case MK379 may not be direct FB inputs to V1→V2 neurons, but rather contact V2 cells sending FB projections to V1.

**Control experiments.** There are two limitations to the TRIO method that need to be addressed: the local TVA "leak", and the possibility of retrograde AAV infection. Both are discussed in the following section.

It has been well-documented in mouse models that small amounts of TVA-mCherry can "leak" out at the injected site and become expressed in cells in the absence of Cre[32,33,40]. Due to the high sensitivity of TVA, this small leak is sufficient to lead to EnvA-RVdG-GFP infection of cells expressing TVA but too low for mCherry to be detected. These RVdG-infected cells can, thus, express GFP in a Cre-independent fashion. This explains why some GFP label was always observable at the injected V1 site, even in the absence of a red label, when the injections were not in good alignment with the AAV and/or CAV2 injections (as for the

lateral and medial injections in Fig. 2a). Importantly, however, TVA leak does not lead to *trans*-synaptic RVdG infection, as the latter requires high levels of oG expression to occur. In order to minimize the amount of local TVA leak, we reduced the concentration of the AAV9-Flex-TVAmCherry virus relative to that of the AAV9-Flex-oG, from 1:1 (in case of MK379) to 3:7 (in the remaining cases; see Supplementary Table 1). To ascertain that the GFP label in V2 and extrastriate cortex described in the Results above was, in fact, Cre-dependent, as well as to determine the amount and extent of local Cre-independent GFP expression in macaque cortex using our protocol, we performed control experiments (n = 2) in which CAV2-Cre was omitted from the TRIO injection protocol described in Fig. 1a. These cases received a single injection of the AAV and RVdG viruses, separated by 3–5 weeks (see Supplementary Table 1), in volumes that were matched to those used for the actual experiments. These controls demonstrated that Cre-independent GFP expression due to TVA leak only occurs nearby the location of the injection site, with 86% of the labeled cells, in one control case, and 80% in the other case, being located within 400 μm of the injected site (Supplementary Fig. 2). Only 16 out of 117 (13.7%) artifactual GFP-labeled cells were located beyond 400 μm of the injection site center, in one case, and only 89 of 449 (19.8%) cells in the second control case (Supplementary Fig. 2c, d). These controls allowed us to establish that all of the inputs arising from outside V1, and the vast majority of the intra-V1 inputs arising from beyond 400 μm from the injection site were dependent on CAV2-Cre, indicating that our approach is well suited to study long-distance inputs. As the local GFP inputs at the injected RVdG injection site is contaminated by artifactual label, we omitted from counts GFP-labeled cells within 400 μm radius of each V1 injection site, in our quantitative analysis of the intra-V1 inputs to V1→V2 cells (described in the next section of the "Results").

A second limitation of the TRIO method is a retrograde infection of neurons by AAV9 vectors, which, although much less

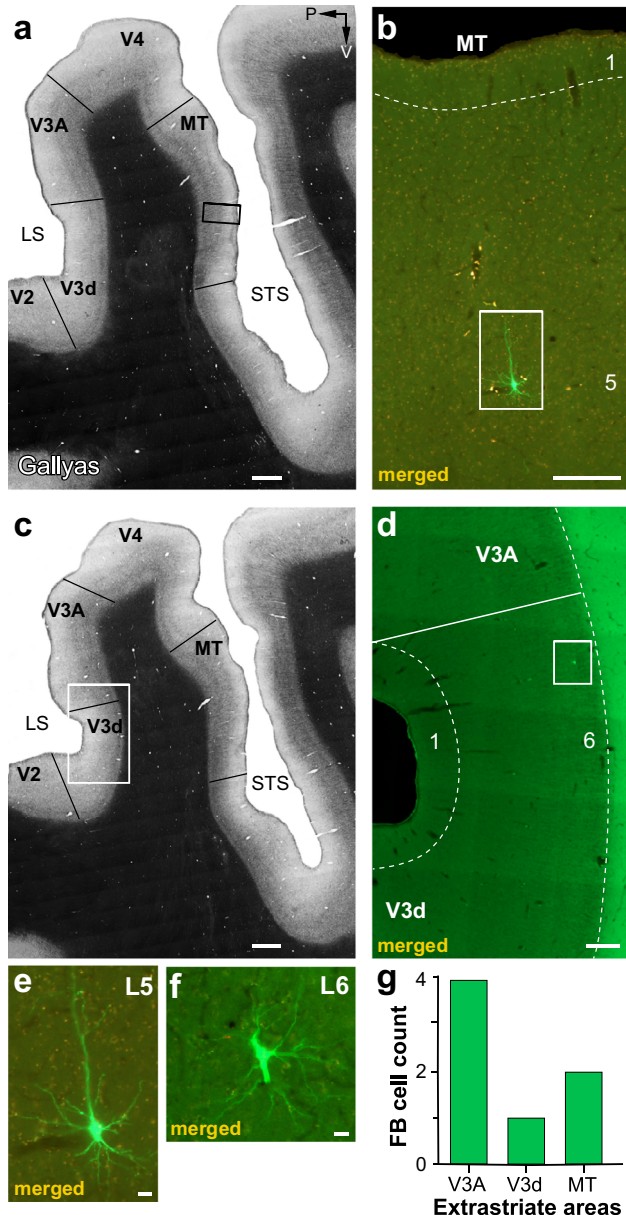

**Fig. 6 Case MK379: FB inputs from the higher extrastriate cortex.**
**a** Image of a sagittal section through the extrastriate cortex encompassing the anterior bank of the lunate sulcus (LS), the prelunate gyrus, and the banks of the superior temporal sulcus (STS), stained for myelin using the Gallyas method to reveal areal borders (solid black lines). P: posterior; V: ventral. **b** Higher magnification of the MT region inside the black box in (**a**) in an adjacent section imaged for GFP and mCherry fluorescence and merged. A single GFP-labeled pyramidal cell is visible in L5 (inside the white box) and shown at higher magnification in (**e**). **c** Same as in (**a**) but for a different section. **d** Higher magnification of the V3d/V3A region inside the white box in (**c**) viewed under fluorescence. A single GFP-labeled cell is visible in L6 of dorsal V3 (V3d) (inside the white box) and shown at higher magnification in (**f**). **e, f** GFP-labeled cells in MT L5 and V3d L6, respectively. Scale bars: 1 mm (**a–c**), 250 μm (**b, d**), 20 μm (**e, f**). **g** Number of GFP-labeled cells in higher extrastriate areas.

efficient than anterograde infection, is known to occur[41]. Thus, V2 neurons sending FB projections to the V1 AAV9 injection sites could potentially be infected by one or both AAV vectors; if these V2 neurons are also retrogradely co-infected by CAV2-Cre, via axon collaterals to the injected V2 site, then TVA-mCherry

and/or oG can be expressed in these cells. Moreover, RV$dG$ can potentially infect these TVA-expressing V2 FB cells at their V1 terminals[42], resulting in double-labeled cells (yellow) in V2. Importantly, only if these cells co-express sufficient levels of oG, can trans-synaptic RV$dG$ infection occur, and lead to GFP expression in their presynaptic partners. In other words, retrograde infection of V2 FB neurons by only three of the vectors (CAV2-Cre, AAV-Flex-TVA-mCherry, and RV$dG$-GFP) is sufficient to double-label these cells, but infection by all four vectors (the three above plus AAV-Flex-oG) must occur for the double-labeled cells to act as starter cells. Lower AAV injection volumes and shorter post-RV$dG$ injection survival times can effectively reduce the efficiency of retrograde AAV infection and trans-synaptic expression of GFP from these retrogradely infected cells[42]. Across our sample ($n = 3$ cases), in the case that received the smaller AAV injection volumes (MK382, Supplementary Table 1) we found double-labeled cells only in V1. In the case that received larger AAV injection volumes but shorter post-RV$dG$ injection survival time (MK405, Supplementary Table 1), 2% of double-labeled cells were found in V2 (of which 75% were in L2/3 and 25% in L5; Supplementary Fig. 3a–d). Finally, in the case that received larger AAV injection volumes and longer post-RV$dG$ injection survival time (MK379, Supplementary Table 1), 17% of the total number of double-labeled cells were located in V2 L5 (Supplementary Fig. 3d). Double-labeled cells in V2 were found at distances up to 1.7 mm from the injection site in V2 (average mean distance ± s.e.m.: 845 ± 174.35 μm; average median ± s.e.m.:758 ± 194.1 μm; Supplementary Fig. 3e); their overall cortical extent along the tangential domain of V2 was 1.9 mm (MK379) and 2.6 mm (MK405), thus smaller than the spread of double-labeled cells in V1 (3.8 mm in MK379, 3.5 mmm in MK405). Consistent with the interpretation that the double-labeled cells in V2 are V2→V1 FB cells sending collaterals to the V2 injected site, and thus retrogradely infected from both V1 and V2, we found no double-labeled cells in V2 layers that are not the source of FB projections to V1, such as L4.

While these double-labeled cells in V2 could represent a second source of starter cells, and thus a potential confound in our study, several lines of evidence suggest that in two of the three cases all, or the vast majority of, the GFP label was presynaptic to the V1→V2 starter cells, while in the third case (MK379) a small fraction of the GFP label in V1 and V2, and all the sparse GFP label outside V1 and V2 was pre-synaptic to the V2 starter cells. This evidence is discussed below. In the case of MK382, which only had double-labeled cells in V1, any GFP label within V1 (>400 μm from the injected site) and outside V1 could unequivocally be interpreted as input to the double-labeled V1→V2 neurons. This case, therefore, demonstrates the existence of area-specific monosynaptic V2 FB inputs to V1→V2 cells. Four observations suggest that in the case of MK405 there was no significant trans-synaptic RV$dG$ infection from the V2 double-labeled cells, perhaps because only a subset of the few double-labeled cells was co-infected by the AAV-Flex-oG vector. First, since cortical neurons receive the majority of their inputs from their neighbors[43], "real" starter cells are expected to be surrounded by GFP-labeled cells, representing their local inputs. Indeed, this was typically observed for starter cells in V1 in all cases (e.g., Fig. 2b), and starter cells in V2 in case MK379, in which every double-labeled cell in V2 was surrounded by GFP-labeled neurons within 150 μm of its location (Supplementary Fig. 4a, c). In contrast, in MK405 only 5 of 11 double-labeled cells in V2 were surrounded by at least one GFP labeled cell within 150 μm distance, (Supplementary Fig. 4b, c). Second, because in case MK405 the majority of the V2 double-labeled cells were located in L2/3 (Supplementary Fig. 3d), and these layers receive most of their local interlaminar inputs from L4[44,45], real starter

cells in V2 L2/3 are expected to produce GFP label in L4. Instead, we found only 3 GFP-labeled cells in V2 L4 (0.4% of total V2 GFP label) in this case (Fig. 5a), suggesting most of the V2 double-labeled cells did not act as starter cells. In contrast, in case MK379, where all V2 double-labeled cells were located in L5, which receives inputs from L4[44], a larger, albeit still small, number of GFP-labeled cells were found in L4 ($n = 20$ cells, 1.7% of the total V2 GFP label; Fig. 5a). Third, in case MK405, the laminar distribution of GFP-labeled cells in V2 was virtually identical to that of case MK382 (Fig. 5a), in which all GFP label was pre-synaptic to the V1 starter cells. In contrast, in the case of MK379, the V2 GFP label was strongly biased to L5 (Fig. 5a), i.e., the layer where all the V2 double-labeled cells resided (Supplementary Fig. 3d); this suggests that at least some of the GFP-labeled cells in L5 in case MK379 were pre-synaptic to the V2 starter cells. Lastly, in the case of MK405, unlike case MK379, we found no GFP label outside V1 and V2. As V2 receives projections from extrastriate areas anterior to it[39], the lack of GFP label in higher extrastriate areas indicates that no *trans*-synaptic RV$d$G infection of long-distance inputs to the V2 double-labeled cells occurred in case MK405. We interpret the GFP-labeled cells in the extrastriate cortex anterior to V2, in the case of MK379, as pre-synaptic inputs to the V2→V1, rather than V1→V2, starter cells. This is because GFP-labeled cells in the higher extrastriate cortex were only found in this case, which had a larger fraction of double-labeled cells in V2. Moreover, the extrastriate areas showing GFP labels, i.e., V3, V3A, and MT, all project to V2[18,29,39], but of these areas only V3 and MT project to V1, thus at least the GFP-labeled cells in V3A must have been pre-synaptic to the V2 starter cells. This suggests the existence of cascading FB-to-FB projections connecting higher areas to V1 via a single synapse within V2.

**V1→V2 neurons receive the majority of monosynaptic cortical inputs from within V1.** In all three cases, we found many GFP-labeled input cells within V1 (e.g., Fig. 7a–c). As discussed above, because the GFP label at the RV$d$G injection sites was contaminated by Cre-independent artifactual label, due to a TVA leak, in our quantitative analyses of the intra-V1 GFP label, we omitted from counts GFP-labeled cells within 400 μm radius of each V1 injection site. Thus our analysis of V1 inputs only included the long-range inputs (>400 μm). In all cases, GFP-label > 400 μm from the injection sites was found in all V1 layers except L1 (Fig. 7d). In each case, the laminar distribution of long-range V1 inputs closely matched the laminar location of the V1→V2 starter cells. Thus, in cases, MK405 and MK379, the majority of V1 inputs were located in L2/3, where most of the V1→V2 starter cells were also located (Fig. 4a), but in case MK382, where L2/3 and 5 had more similar percentages of starter cells (Fig. 4a), similar amounts of labeled intra-V1 horizontal inputs were found in both layers. On average across the population, 46.9 ± 7.3% of V1 horizontal inputs arose from L2/3, followed by L5 (23.7 ± 7.5%) (Fig. 7e). This is consistent with the well-known prominence of intra-laminar horizontal connections in V1 L2/3 and L5[46,47]. Our counts of GFP labeled intra-V1 long-range inputs included some cells labeled artifactually as a result of TVA leak, as our control experiments demonstrated that 14–20% (14–89 cells) of artifactual GFP label occurred at distances >400 μm from the injected RV$d$G sites (see above and Supplementary Fig. 2). However, given the large numbers of GFP-labeled cells counted in V1 at >400 μm distances (range: 2569–10,688 cells) the potential inclusion of 14–89 artifactual cells to these counts is negligible.

The maximum tangential spread of the V1 long-range inputs ranged between 6.4 and 10.3 mm (mean max spread ± s.e.m: 8.73 ± 1.17 mm). After eliminating the tails of the distributions, this

range was 2.8–5 mm (mean ± s.e.m.: 3.8 ± 0.65 mm) (Fig. 7f). For each case, we calculated a ratio of the spatial spread of the V1 inputs to that of the V1→V2 starter cells. This ratio ranged between 2.5 and 3.6 (mean ± s.e.m: 3.03 ± 0.32) or, after removing the tails of the distributions, 1.4 and 1.8 (mean ± s.e.m: 1.6 ± 0.13). Thus, the spread of V1 inputs is about 1.6–3 times the size of the V1 region to which they project. The ratio of the V2 FB spread to that of the V1 input spread averaged 1.1 ± 0.13 (or 1.5 ± 0.06 after removing the tails), indicating the cortical spread of long-range intra-V1 inputs was only slightly less extensive than the spread of V2 FB inputs to the same cells. However, when considering the larger RFs and lower magnification factor in V2 compared to V1, the visuotopic extent of V2 FB is larger than that of intra-V1 inputs (see "Discussion"). These results are consistent with previous reports on the relative spatial spread of V1 horizontal connections and FB connections from V2 to V1[38].

On average across the population, V1→V2 starter cells received 91.6 ± 3.1% of their total long-range cortical mono-synaptic inputs from other V1 cells, with only 8.4 ± 3.1% arising from V2 (Fig. 7g). For the different cases, however, this percentage varied with the percent of V1 versus V2 starter cells. Specifically, in cases MK405 and MK382, where 98% and 100%, respectively, of starter cells, were located in V1, 93% and 96%, respectively, of their monosynaptic inputs arose from within V1. Instead, in case MK379, where 83% of starter cells were located in V1 and 17% in V2, a lower percent (85%) of GFP-labeled inputs were located in V1, supporting our interpretation above that some fraction of the GFP-labeled V2 input cells, in this case, were presynaptic to the V2 starter cells rather than to the V1→V2 cells. Overall, these results indicate that monosynaptic FB inputs to V1→V2 neurons are sparse.

**Monosynaptic inputs from the thalamus.** V1 receives sub-cortical inputs from the lateral geniculate nucleus (LGN) of the thalamus[48,49], as well as the pulvinar[50], a higher-order thalamic nucleus. We asked whether any of these inputs make direct synaptic contacts with V1→V2 cells. In two cases, MK405 and MK382, we found no GFP-labeled cells in either the LGN or the pulvinar, suggesting that thalamic inputs undergo intra-V1 processing before being relayed to V1 corticocortical output cells. In contrast, in the case of MK379, we found a small percent of GFP-labeled input cells (0.22% of total) in the LGN ($n = 14$ cells) and pulvinar ($n = 4$ cells; Fig. 8). In the LGN, 86% of input cells were found in the parvocellular (Parvo) layers, 14% in the magnocellular (Magno) layers, and none in the koniocellular (Konio) layers (Fig. 8a–c, g). As input cells in the LGN were found only in this case, and this is the only case that had a significant fraction of "real" starter cells in V2, and because V2 is known to receive a small number of direct inputs from LGN[51], it is likely that the GFP-labeled cells in the LGN, in this case, represent direct monosynaptic geniculate inputs to the V2→V1 starter cells. This would suggest the existence of direct geniculate inputs onto V2 cells sending FB connections to V1. Because extrastriate geniculate inputs have been shown to arise primarily, albeit not exclusively, from the Konio layers[51–53], we immunoreacted the LGN for calbindin, a neurochemical marker of the Konio geniculate channel[54]. None of the GFP-labeled cells co-expressed calbindin, suggesting they may not be Konio cells (Fig. 8a–c). We found only four GFP-labeled cells in the lateral subdivision of the pulvinar (Fig. 8d–f), the latter identified as a region of sparser calbindin expression compared to its neighboring inferior subdivision[55]. Based on the same rationale as for the GFP-labeled LGN cells, it is likely that the GFP-labeled cells found in the pulvinar, in this case, represent direct inputs to the starter cells in V2 that send FB to V1.

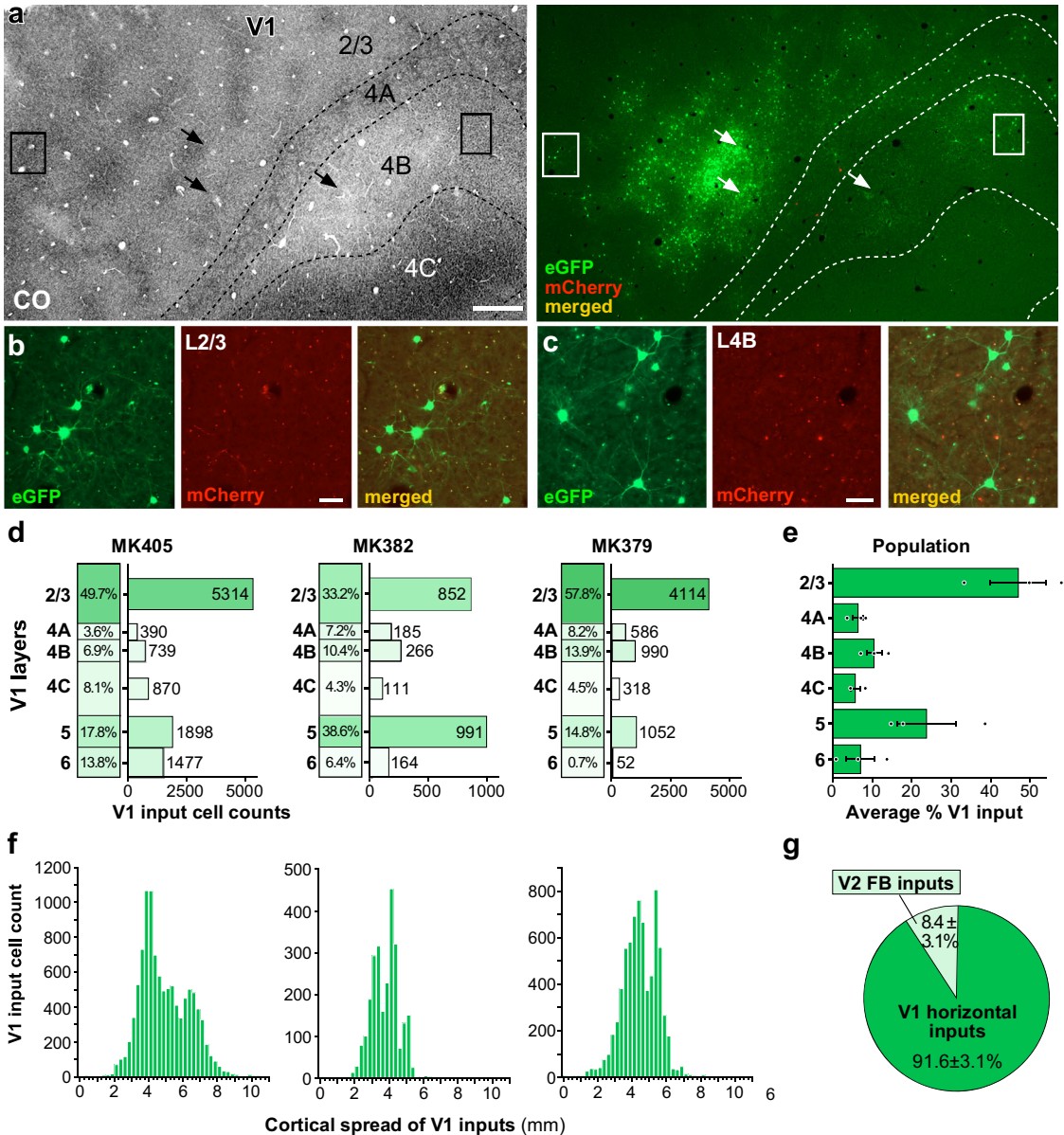

**Fig. 7 Laminar and tangential distribution of long-range V1 inputs. a** Case MK405. Image of a single tangential section through V1 L2/3–4C stained for CO (Left) after being imaged for mCherry and GFP fluorescence (Right). The merged channel shows plenty of GFP-labeled V1 input cells (green) in L2/3, 4A–4C away from the injection sites (arrows). The locations of the injection sites were determined in more superficial sections where the CO discoloration was more visible than in the indicated section. The region inside the boxes is shown at higher magnification in (**b**, **c**). Other conventions are as in Fig. 2. Scale bar: 500 μm. **b**, **c** GFP-labeled V1 input cells in L2/3 (**b**) and L4B (**c**), shown under GFP (Left) or mCherry (Middle) fluorescence, and merged (Right). Scale bars: 50 μm. Results in (**a**–**c**) are representative of the V1 GFP label in three independent TRIO experiments. **d** Percent and number of long-range V1 input cells across layers for each of the three cases. **e** Average percent ± s.e.m. of V1 input cells across V1 layers for the population. **f** Tangential spread of V1 input cells for each case (GFP-labeled cells within 400 μm of each V1 injection site were omitted from the counts). **g** Average percent of cortical inputs arising from V2 vs. V1 for the population (n = 20,369 cells in 3 independent animals). Error bars: s.e.m. Results in (**a**–**c**) are representative of injection sites made in three independent cases with similar results.

## Discussion

Using TRIO labeling in the macaque visual cortex, we have demonstrated the existence of area-specific monosynaptic FB contacts with FF-projection neurons. Specifically, we have shown that

V1 neurons sending FF projections to V2 receive direct monosynaptic inputs from V2 FB neurons, but not from neurons in other extrastriate areas known to project to V1. FB-to-FF inputs occur in both superficial and deep V1 layers, although our approach did not allow us to determine the differential contribution of superficial and deep layer FB neurons to the V1

termination layers. These direct interareal FB inputs represent only a tiny fraction of the total long-range cortical inputs to V1 cortical projection neurons, which overwhelmingly arise from within V1. We also found evidence for the existence of direct monosynaptic interareal FB-to-FB contacts relaying top-down information from higher extrastriate areas to V1, via a single synapse in V2. Finally, we found sparse direct inputs from the Parvo and Magno LGN layers and lateral pulvinar to V2 L5 neurons sending FB projections to V1 (Fig. 9).

It is well established that in the primate visual cortex, most V1 cortical projection neurons send FF projections to only a single

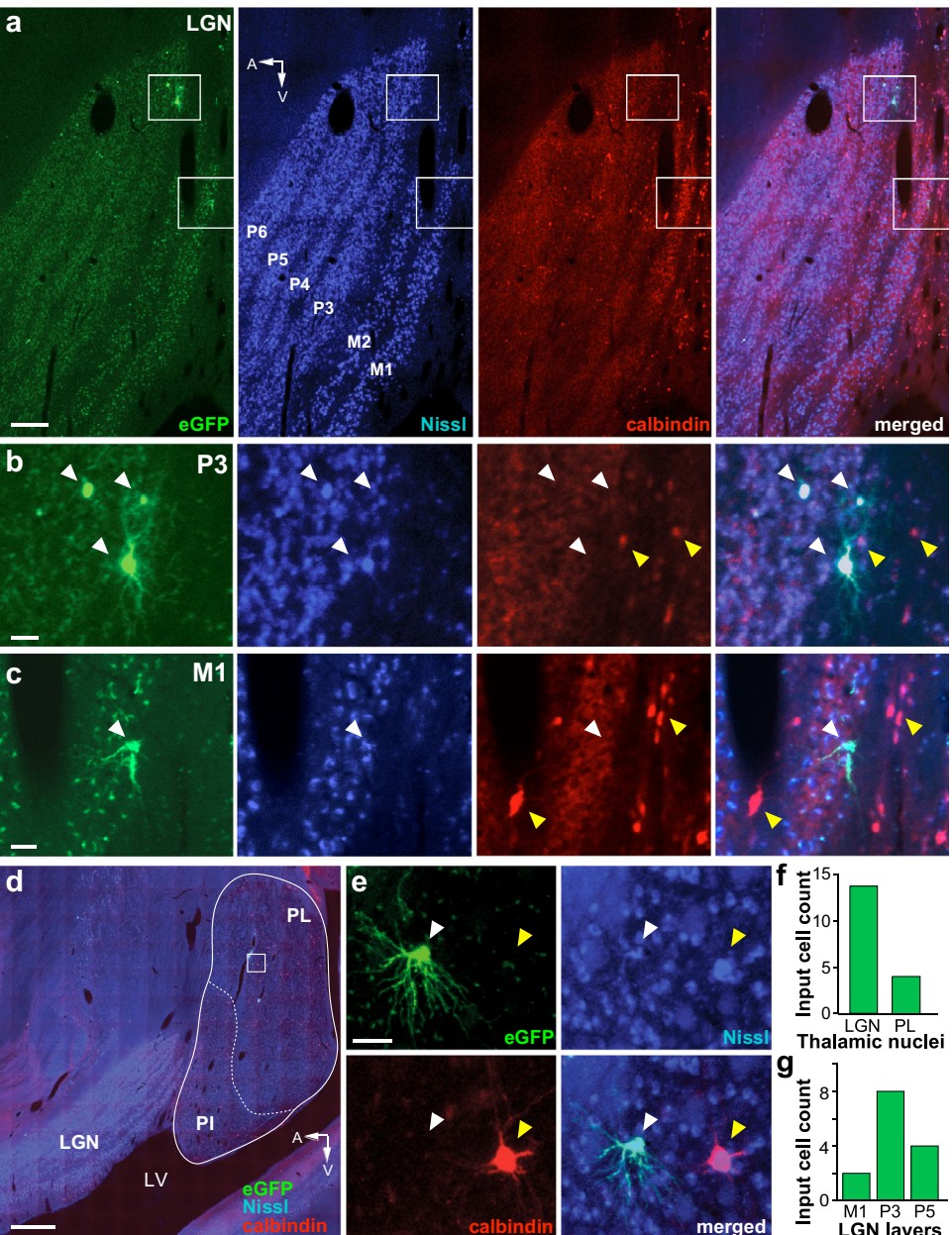

**Fig. 8 Thalamic input cells. a** Case MK379. Image of a single parasagittal section through the LGN viewed under GFP fluorescence (Left), stained for fluorescent Nissl (Middle Left), immunostained for Calbindin-Alexa647 (Middle Right), and with all three channels merged (Right). The GFP-labeled cells inside the top and bottom white boxes are shown at higher magnification in (**b**, **c**), respectively. The parvocellular (P3–6) and magnocellular (M1–2) LGN layers are labeled. A: anterior; V: ventral. Scale bar: 250 µm. **b**, **c** GFP-labeled LGN input cells in the P3 (**b**) and M1 (**c**) layers are shown in the same three channels as the top panels and with all channels merged (Right). White arrowheads point to the location of GFP-labeled neurons, yellow arrowheads point to calbindin-positive cells. The GFP-labeled cells are not calbindin-positive. Scale bars: 50 µm. **d** Image of a sagittal section through the LGN and pulvinar, with all three fluorescent channels (GFP, calbindin, and Nissl) merged. The cells inside the white box are shown at higher magnification in (**e**). PL lateral pulvinar, PI inferior pulvinar, LV lateral ventricle. Scale bar: 1 mm. **e** A GFP-labeled input cell (white arrowhead) in the PL imaged under the same three channels as for the LGN cells and with all channels merged (Bottom Right). A yellow arrowhead in each panel points to the location of a calbindin-positive cell (red). The GFP-labeled cell is not calbindin-positive. Scale bar: 50 µm. **f** Number of GFP-labeled cells in the thalamic nuclei. **g** Number of GFP-labeled cells in the LGN layers.

area[21–23], but it was unclear whether FB is also area-specific and whether it influences FF afferents directly or indirectly. Our results demonstrate that in primate V1, FF-projection neurons receive direct monosynaptic FB inputs selectively from the area to which they project. We found monosynaptic inputs to V1→V2 neurons from V2, but inputs from other extrastriate areas known to project to V1 were either absent or extremely sparse. Henrich

et al.[56] recently reported that the probability of rabies virus *trans*-synaptic spread at each synapse is about 30%; thus, the number of labeled input neurons increases with the number of synapses formed by presynaptic neurons onto a given starter cell, providing an indirect measure of the functional strength of a projection. Thus, the absence of labeled presynaptic neurons in the extrastriate cortex anterior to V2 strongly suggests that such

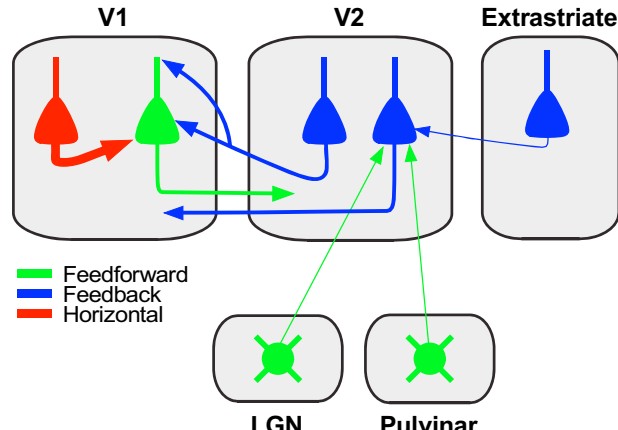

**Fig. 9 Summary circuit model. Schematics of the FB circuit motifs were discovered in this study.** Triangles: pyramidal cell somata; circles: thalamic cell somata; arrows: axonal projections (thickness indicates projection magnitude). All axonal projections in this scheme are excitatory and terminate onto excitatory cells. Some V2 FB neurons (left V2 blue cell) make monosynaptic contacts with V1 neurons projecting to V2 (green pyramidal cell). The latter receive the majority of their long-range cortical inputs from other pyramidal neurons within V1 (red cell). Some V2 neurons in L5 (right V2 blue cell) sending FB to V1 receive monosynaptic inputs from FB neurons in higher extrastriate areas (blue cell in the extrastriate cortex), as well as sparse inputs from the LGN and lateral pulvinar (round green cells).

projections are either absent or so much sparser than the projection from V2 that our method failed to reveal them. These findings support the existence of highly area-specific FB-to-FF contacts in primate V1. This is in contrast with results from mouse V1, where about 80–88% of FF projection neurons project to one or two higher visual areas[19,57], but only about 50% of their monosynaptic FB contacts arise from the same areas to which they project[19]. Moreover, recent evidence suggests that in the mouse a bias to form area-specific monosynaptic excitatory FF–FB loops may be limited to deep layer neurons[20], while our results in macaque demonstrate area-selective FB-to-FF contacts in both V1 superficial and deep layers. These findings strongly support the existence of area-, and thus, functionally-specific, FF–FB loops in the primate cortex.

Several previous studies have shown, albeit mostly qualitatively, that the V2 deep layer FB projections to V1 arise predominantly from L6, and less so L5[18,25,29,39]. In contrast, here we found the vast majority of deep layer FB to arise from L5. Even allowing for the possibility of some errors in the laminar assignments of cells located near the L5/6 border, given the tangential sectioning plane used in our study, the bulk of the V2 deep layer GFP label clearly lay in L5. However, unlike previous studies which labeled all V2 FB projections to V1, we selectively labeled the FB projections to V1→V2 cells. Thus, our findings suggest laminar specialization in the deep layer FB projections, with FB from L5 contacting preferentially V1→V2 neurons.

Our findings support several theories advocating looped computations between FF and FB connections[8–16]. The specific nature of the computations performed by these loops varies with the specific theory. For example, in predictive coding theory, FB signals represent a prediction of the external world, based on sensory data and prior experience; this prediction is compared with incoming sensory data, and the prediction error, carried by FF-projecting "error units", ascends up the cortical hierarchy and refines the higher level predictions[8,9,58]. In terms of their architecture, predictive coding schemes require both excitatory and

inhibitory looped interactions of FB inputs with lower-level "error units" to signal mismatches between predictions and sensory inputs[59,60]. Importantly, these looped interactions must be area-specific and occur between FF and FB units encoding similar features. Our findings of area-specific monosynaptic FB contacts with FF-projecting neurons, together with recent evidence of stream-specific V2 FB projections to V1[30], support the area and functional specificity of FF–FB loops required by predictive coding theories. Moreover, these FB-to-FF contacts could provide an anatomical substrate for excitatory FB interactions with the lower-level error units (so-called "negative error units") required by predictive coding[60]. Alternatively, direct FB-to-FF contacts could underlie the "precision" FB signals of predictive coding models[59,60]. In the latter, the precision FB circuit is distinct from the prediction circuit and provides a modulatory or gating FB signal that sets the weight or precision of the prediction error. Whether the direct FB-to-FF contacts we have found serve to compute prediction errors or modulate their precision, ultimately depends on whether such contacts occur on the basal or apical dendrites, respectively, of V1→V2 cells[59]. Both kinds of FB contacts can potentially occur, the former in L5/6, and the latter in L1/2, where FB connections mostly terminate[30]. However, our method did not allow us to determine in which V1 layers these monosynaptic FB contacts occur.

An additional key component of predictive coding models is the inhibitory FB interaction with the lower-level error units (so-called "positive error units"), which requires FB contacts with inhibitory neurons. Moreover, experimentally FB has been shown to cause both facilitation and suppression of neural activity in lower-level areas[61–63]. Thus, the direct FB-to-FF connections we have found here could underlie the facilitatory effects of FB, but direct or indirect contacts with inhibitory neurons are necessary to mediate the FB suppressive effects found experimentally and postulated by predictive coding. Indeed, direct FB contacts with inhibitory neurons have been demonstrated in both mouse[64–66] and primate[67] visual cortex. Therefore, the monosynaptic FB–FF contacts we have found in this study represent just one of several motifs of FB connectivity in the primate cortex.

Our approach did not allow us to determine whether the V1→V2 neurons receiving the direct FB contacts directly target the same V2 neurons that are the source of their FB input. Notably, these direct FF–FB contacts are not required by predictive coding schemes. On the contrary, several of the proposed schemes view FF inputs from lower-level error units indirectly affecting their looped prediction FB units, via contacts with local neurons making recurrent connections with each other[59,68]. This intra-areal recurrent processing between local expectation units serves to generate, maintain, and refine the internal predictions, which are then passed on to the FB units for relay to lower-level areas. Moreover, the termination of FF pathways from V1 predominantly in L4 and lower 3 of V2, and the origin of V2 FB pathways in layers 2/3 and 5/6[29] would make direct FF-to-FB contacts less probable than indirect ones.

A different theory postulates that fast recurrent FB–FF processing between adjacent hierarchically organized areas serves to facilitate object recognition, particularly when incoming sensory inputs are ambiguous, degraded, or noisy[12]. In these models, these local FB signals are fast, operating during the initial FF process. The area-specific monosynaptic FB-to-FF contacts we have found here represent an ideal anatomical substrate for fast and specific facilitatory FB modulation of incoming FF signals, and are also consistent with evidence that FB acts on the early part of the FF-driven response[69].

We found that FB inputs to V1→V2 neurons arise from a cortical region extending on average 9.9 mm, approximately 2–3 times larger than the size of their V1 target zone, and similar in

cortical extent to the spread of long-range intra-V1 horizontal inputs (8.7 mm; 1.5–3 times the size of their target V1 zone). These results are qualitatively consistent with previous reports[38] that V2 FB connections to a V1 column extend on average 6.4 mm (reaching up to 9.4 mm), and are similar to the average cortical extent of V1 horizontal connections to the same column (mean 7.9 mm, max 9.5 mm). However, the same study demonstrated that, due to the larger RF sizes and lower magnification factor in V2 compared to V1, the visuotopic extent of V2 FB connections is larger than that of V1 horizontal connections to the same V1 column. Specifically, while V2 FB connections convey information from a visual field region about 4–6 times the size of the aggregate RFs of their target V1 cells, the visuotopic extent of horizontal connections is only 2–4 times the aggregate RF size of their target V1 cells[38]. Qualitatively similar results were recently reported for FB connections in mouse visual cortex[70], suggesting that a feature of FB connections conserved across species is their ability to convey information from distant visual field locations to their postsynaptic neuronal targets. This feature has been proposed to underlie contextual modulations from outside the classical receptive field[71].

Our results demonstrate that FF-projecting V1 neurons only receive a small fraction of their direct long-range (>400 μm) cortical inputs from FB neurons, the majority of which, instead, arise from neurons within V1, particularly in L2/3 and L5, where intralaminar horizontal connections are known to be most prominent[46]. While it is well established that cortical neurons receive the majority (79%) of their inputs from neurons within the same cortical area[43], this is the first demonstration in the primate cortex that this connectivity rule also applies specifically to cortical projection neurons.

At least in the dorsal stream of visual processing, we found evidence for the second motif of FB connectivity, namely inter-areal FB-to-FB neuron contacts (Fig. 9). These chains of FB connections may serve to convey fast FB modulations, possibly related to the processing of object motion, from higher cortical areas V3, V3A, and MT to V1, via V2. Alternatively, given that area V3 and MT also send direct inputs to V1, these FB inputs to V2 may serve to specifically modulate V2 FB inputs to V1. These FB-to-FB circuits represent a sparse projection, as we only found a total of 7 neurons in higher extrastriate areas potentially projecting to 11 L5 V2→V1 cells.

The same V2→V1 FB cells that received direct FB inputs from higher extrastriate areas of the dorsal stream also received a small direct FF input from the LGN and lateral pulvinar (Fig. 9). We found GFP-labeled neurons in the LGN and pulvinar only in the one case that showed labeled starter cells in V2 L5, suggesting these thalamic inputs target, and thus can directly influence the activity of, these V2→V1 L5 FB neurons. It is well-known that V2 receives a sparse direct projection from the LGN, which has been postulated to be part of a retino-colliculo-thalamic pathway to extrastriate cortex[43,51]. However, while this projection, as well as direct geniculate projections to other extrastriate areas, arise predominantly from calbindin-positive or CaMKII-positive Konio geniculate neurons terminating in L4 and 5[51–53], here we find these direct LGN-to-V2 FB contacts arise from calbindin-negative cells in the Parvo and Magno LGN layers. While lack of calbindin immunoreactivity suggests neurons giving rise to these projections may not belong to the Konio system, it has been noted that this system is heterogeneous and also includes neurons that are calbindin and CaMKII-negative[52,53]. We cannot exclude that at least some of the Parvo LGN inputs were, instead, presynaptic to the starter V1 cells in L4A, as Parvo-to-L4A projections exist[72]; this would indicate the existence of direct geniculate inputs to L4A output cells. However, we believe this is unlikely, because two of our cases with starter cells in L4A, but few or no

starter cells in V2, showed no labeled input cells in the LGN. Similarly, as the lateral pulvinar in addition to V2 also projects to L1–2 of V1[44,50], it is possible, although unlikely, that the pulvinar inputs we observed here were instead presynaptic to the apical dendrites of the starter cells in V1 L2-4B or L5. This would suggest that the pulvinar can directly affect the activity of V1 cortical output cells. Finally, as Magno afferents only terminate in V1 L4C and 6, and there were no starter cells in these V1 layers, it is unlikely the sparse Magno inputs found in our study represent direct inputs to V1 output cells.

## Methods

**Experimental design.** We performed monosynaptic input tracing or TRIO to label monosynaptic inputs to V1→V2 neurons (starter cells) in the macaque monkey visual cortex. The method consisted of targeting injections of three different viral vectors to V1 or V2, identified in vivo by intrinsic signal OI. The resulting labeled starter cells and input cells were mapped throughout V1, V2, extrastriate cortex, and thalamus, and their laminar and tangential distributions were analyzed quantitatively.

**Animals.** We made a total of 25 viral injections in five adults (3–5 years old) female cynomolgus macaque monkeys (*Macaca fascicularis*). Three animals were used for regular TRIO experiments and two for control experiments. All procedures involving animals were approved by the Institutional Animal Care and Use Committee of the University of Utah and conformed to the guidelines set forth by the USDA and NIH.

**Surgical procedures.** Animals were pre-anesthetized with ketamine (10 mg/kg, i. m.). An i.v. the catheter was inserted, and the animals were intubated with an endotracheal tube, placed in a stereotaxic apparatus, and artificially ventilated. Anesthesia was maintained with isoflurane (1–2.5%) in 100% oxygen, and end-tidal $CO_2$, blood oxygenation level, electrocardiogram, and body temperature were monitored continuously. I.V. fluids were delivered at a rate of 5/cc/kg/h. The scalp was incised, a large craniotomy and durotomy (about 15–20 mm mediolaterally and 6–8 mm anteroposteriorly) were made to expose the lunate sulcus, area V2 and parts of V1 (e.g., Fig. 1b). A clear sterile silicone artificial dura was placed on the cortex, and the craniotomy was filled with a sterile 3% agar solution and sealed with a glass coverslip glued to the skull with Glutures (Abbott Laboratories, Lake Bluff, IL). On completion of the surgery, isoflurane was turned off and anesthesia was maintained with sufentanil citrate (5–10 μg/kg/h, i.v.). The pupils were dilated with a short-acting topical mydriatic agent (tropicamide), the corneas were protected with gas-permeable contact lenses, the eyes refracted, and OI was started (see below). Once the V1/V2 border was functionally identified (1–4 h. of imaging), the glass coverslip, agar, and artificial dura were removed, and 2–3 injections of AAV9 vectors (see below) were made in V1 using surface blood vessels as guidance. On completion of the injections, new artificial dura was placed on the cortex, the native dura was sutured over the artificial dura, the craniotomy was filled with Gelfoam and sealed with sterile parafilm and dental cement, the skin was sutured, and the animal was recovered from anesthesia. Animals survived 21–24 days post-injections, and then we're prepared for a second surgical procedure and anesthetized with isoflurane as described above. The scalp was re-incised at the same site as the prior incision, the artificial dura was removed, and 1–2 injections of the CAV2 vector (see below) were made in V2, using as guidance the surface blood vessels and functional OI maps obtained in the first surgical procedure (Fig. 1b). The animals were recovered as described above, and after a 2-11-day survival time underwent a third surgical procedure during which multiple injections of the RVdG vector (see below) were made at the same locations as the previously made AAV injections in V1, again using blood vessels as guidance. Animals survived an additional 9–12 days, during which a terminal 2–3 day OI experiment was performed to obtain additional functional maps. At the conclusion of the OI experiment the animal was sacrificed with Beuthanasia (0.22 ml/kg, i.v.) and perfused transcardially with saline for 2–3 min, followed by 4% paraformaldehyde (PFA) in 0.1 M phosphate buffer for 20 min.

**Optical imaging.** Acquisition of intrinsic signals was performed under red light illumination (630 nm) during the first survival surgery and, then, again during a terminal procedure, using the Imager 3001 and VDAQ software (Optical Imaging Ltd., Israel). We imaged for orientation and retinotopy, as these functional maps allow identification of the V1/V2 border. Orientation maps were obtained by presenting full-field, high contrast (100%), pseudorandomized, achromatic drifting square-wave gratings of eight different orientations at 1.0–2.0 cycles/° spatial frequency, moving back and forth at 1.5 or 2°/s in directions perpendicular to the grating orientation. Responses to the same orientations were averaged across trials, following baseline correction, and difference images were obtained by subtracting the responses to two orthogonally oriented pairs. Retinotopic maps were obtained by subtracting responses to monocularly presented oriented gratings occupying complementary adjacent strips of visual space, i.e., masked by

0.5–1° gray strips repeating every 1–2°, with the mask reversing in position in alternate trials. Baseline correction for both the orientation and retinotopic maps was performed in three different ways and the approach that provided the best maps was selected for analysis: (1) the baseline (pre-stimulus) was subtracted from the single condition response (i.e., the images recorded during stimulation of one stimulus orientation); (2) the single condition response was divided by the baseline; (3) the single condition response was divided by the "cocktail blank" (i.e., the average of responses to all oriented stimuli or all retinotopic stimuli)[73,74]. The V1/V2 border can be identified in the retinotopic maps by the presence of stripes of activity in V1, which are absent in V2 (using the specific stimulus parameters indicated above, which are optimized for V1, but not V2). V2 can be identified in the orientation maps by larger orientation domains compared to V1, and the characteristic "stripy" pattern of orientation domains (e.g., Fig. 1b right). In each case, reference images of the surface vasculature were taken under green light (546 nm) illumination, and used in vivo as a reference to position pipettes for viral vector injections (e.g., Figure 1b Left), and post-mortem to align the in vivo maps with histological tissue sections.

**Injection of viral vectors**. For TRIO experiments, we made a total of 21 injections of 4 different viral constructs in 3 macaques (MK379, MK382, and MK405). The viral vectors were: AAV9-CAG-FLEX-TVAmCherry (titer: $4.69 \times 10^{13}$ GC/ml; Salk Institute Viral Core GT3), AAV9-CAG-FLEX-oG-WPRE (titer: $3.52 \times 10^{13}$ GC/ml; Salk Institute Viral Core GT3), E1-deleted-CAV2-CMV-Cre-SV40polyA (titer: $4.6 \times 10^{12}$ pp/ml; Montpellier Vector Platform, CNRS, France) and EnvA-RV$d$G-eGFP (titer range: $4.69 \times 10^7$–$5.45 \times 10^8$ TU/ml; Salk Institute Viral Core GT3). All viruses were slowly pressure injected at a rate of 6–15 nl/min, using a picospritzer (World Precision Instruments, FL, USA) and glass micropipettes (25–50 μm inner diameter). The two AAV9 vectors were mixed at 1:1 or 3:7 ratio and loaded into the same glass micropipette, and 2–3 injections of the mixture were made into V1, 1–1.3 mm posterior to the V1/V2 border and spaced mediolaterally (i.e., in a row parallel to the V1/V2 border) 1–1.1 mm apart (Fig. 1b Left). These injections were aimed at involving all V1 layers by pressure ejecting half of the total volume at a cortical depth of 800–1200 μm from the pial surface and, after a 5–10 min pause, retracting the pipette to a depth of 400–600 μm and ejecting the remaining volume. The pipette was left in place for an additional 5–10 min before being retracted from the brain, to avoid backflow of solution. After about 21 days, 1–2 injections of the CAV2 vector were made into V2, 1–1.1 mm anterior to the V1/V2 border and, when 2 injections were made, they were spaced 200–300 μm anteroposteriorly (Fig. 1b Left). V2 injections were made as described above for the V1 injections but were aimed at cortical L4–6 (depths 700 and 1000 μm). After 2–11 days of survival, 2–3 injections of the RV$d$G vector were made into V1 at the same locations and depths as the previously made AAV injections, whose location relative to the surface vasculature had been recorded onto the in vivo images obtained during the first surgery. The larger number of injections in V1 allowed us to achieve a larger coverage with the AAV and RV$d$G vectors, to ensure that at least one of these injections was retinotopically matched to the location of the V2 injection site. Survival times were optimized to achieve maximal expression of each vector in the primate cortex while minimizing its potential toxicity. Injection parameters (volumes, numbers, and depths) and inter-injection survival times for each animal are reported in Supplementary Table 1.

**Control injection cases**. A total of four viral injections were made in two additional animals (MK380 and MK381) for control experiments, to determine the amount and extent of Cre-independent GFP expression caused by TVA leak. In each animal, one injection of each of the AAV9 and RV$d$G vectors were made in the motor cortex, using the same viral constructs, injection parameters, depth locations, and survival times (Supplementary Table 1) as used for the regular TRIO experiments, but in these control experiments, the CAV2-Cre injection was omitted.

**Histology**. Areas V1 and V2 were separated from the rest of the visual cortex, by making a cut along the fundus of the lunate sulcus. The block was post-fixed for 1 h in 4% PFA between glass slides, to gently flatten the cortex parallel to the imaged area, cryoprotected in 30% sucrose, and frozen-sectioned at 40 μm on a sliding microtome in a plane tangential to the imaged surface of V1 and V2. Sections were wet-mounted, scrutinized for fluorescent label and, selected sections (two of every three sections) containing label were imaged for fluorescent GFP and mCherry label. After imaging, every third section was reacted on the glass slide for CO, to reveal layers and areal boundaries, and the sections were re-imaged under bright field illumination.

The remainder of the brain, with the frontal pole removed, was post-fixed overnight in 4% PFA, cryoprotected, and sectioned sagittally at 70 μm. A full series of sections was wet-mounted and imaged for the fluorescent labels. Sections containing fluorescent labels were stained for myelin using the Gallyas method[75], to aid in the identification of extrastriate areas and areal boundaries, and stained for fluorescent Nissl to identify cortical layers and subcortical nuclei. Furthermore, to identify the pulvinar subdivisions and the koniocellular layers of the LGN, selected sections containing fluorescent labels were immunoreacted for Calbindin as follows. Sections were incubated in primary antibody (1:5000 monoclonal mouse

anti-Calbindin D-28k; Swant, Switzerland) for 72 h at 4 °C, and then reacted with a secondary antibody tagged to a near-infrared fluorophore (1:200 donkey anti-mouse IgG-AF647; Jackson ImmunoResearch, PA, USA).

**Data analysis**

*Label imaging.* We searched for the fluorescent labels in every section throughout the cortex (except for the prefrontal cortex) and thalamus. We then imaged at regular intervals (two of every three sections) two full series of sections throughout the regions containing labeled cells. Tissue sections were simultaneously imaged for both GFP and mCherry fluorescence, and the sections immunoreacted for Calbindin and stained for Nissl were additionally imaged for Alexa 647 and DAPI, respectively. Imaging was performed on a Zeiss Axio Imager.Z2 fluorescent microscope connected to an Apotome 2, using a 10× objective and an Axiocam 506 mono camera (Zeiss, Germany). Image files were created and analyzed using Zen 2.6 Blue Software (Zeiss, Germany). Sections were imaged using uniform camera exposure times and LED intensity. Imaged sections were scrutinized for fluorescent labels and the regions containing double-labeled (green and red) starter cells were re-imaged at higher magnification, using a 20× objective and the Apotome to obtain z-stacks in 1–2 μm z-steps, so as to verify cells classified as double-labeled. CO and Gallyas stainings were imaged under bright field illumination, using the same microscope and a 10× objective. All images were post-processed in Zen using the "stitching algorithm" to align individual image tiles and minimize tiling artifacts. Images used for figures were exported directly from Zen files, and brightness or contrast was uniformly increased or decreased in Adobe Photoshop across the entire image in each channel.

*Cell counts.* Imaged sections were aligned in a sequential stack through the depth of the cortex (for tangentially sectioned V1/V2 blocks), or in a mediolateral stack (for sagittally-sectioned tissue) using Adobe Photoshop, by registering the radial blood vessels. The aligned image stacks were then transferred to Neurolucida Software (Microbrightfield Bioscience, VT, USA) for cell plotting and counting, and for drawing layer boundaries based on CO and Nissl stains. For each section containing the label, we plotted and counted all GFP-labeled (green) input cells (excluding only GFP-labeled cells in V1 located at distances <400 μm from each injection site center) and all double-labeled (yellow) cells. For V1 and V2, we imaged and counted two full series of sections, while for the thalamus and the rest of the cortex we counted every labeled cell in every section that contained the label. We defined "input" cells as cells exclusively labeled with GFP showing morphological and size characteristics of neurons. Double-labeled cells were defined as somata expressing both GFP and mCherry in the same imaging z-plane, with the two labels perfectly overlapped. Since GFP and TVAmCherry are differently distributed inside neurons, the former filling the soma and the latter binding to the cell membrane, we allowed for the possibility that GFP and mCherry-labeled cells did not show identical shapes. Each cell was additionally assigned a layer location, and the number of cells in each layer, as well as the percentage of total GFP or double-labeled cell counts in each layer, were determined (Figs. 4a, 5a, 7d, Supplementary Fig. 3d). We then averaged these percentages across the 3 cases for each layer and estimated the s.e.m. (Figs. 4b, 5b, and 7e). For each case, we also calculated a ratio of the number of GFP-labeled V2 input cells in each layer to the total number of V1 double-labeled cells (across all V1 layers), and then averaged these ratios across cases by V2 layer (Fig. 5d). We divided by the total number of V1 double-labeled cells, as our method does not allow us to determine to which layers the V2 FB cells project. Finally, for each case, we estimated the percent of total GFP-labeled V1 and V2 input cells that arose from V2 vs. V1 and then averaged those values across cases (Fig. 7g). Plots of cell counts and population statistics were generated in RStudio 1.4.1103.

*The spatial extent of the label.* We quantified the spatial spread of GFP-labeled cells within the tangential domain of V1 (Fig. 7f). This was done by counting GFP-labeled neurons within 250 μm bins along an axis parallel to the V1/V2 border, encompassing the full tangential extent of the GFP label, and excluding GFP neurons within 400 μm of an RV$d$G injection site. We pooled data across layers and injection sites because our method did not allow us to determine to which starter cell each input cell projects. In a similar fashion, we measured the spread of the GFP-label in the tangential domain of V2 along an axis parallel to the V1/V2 border (Fig. 5c), as well as the overall spread of the double-labeled cells in V1 (Fig. 4c) and V2. Spatial spread data was plotted (using RStudio) as histograms of the number of cells as a function of cortical extent, with zero on the x-axis indicating the most medial location of the labeled field and the largest number the distance of the most lateral location of the labeled field from its medial edge (zero). For the two control cases, instead, we measured the distance of each GFP-labeled cell from the center of the injection site (Supplementary Fig. 2c, d). The injection site has identified points of damage or discolorations in CO staining, visible across multiple serial sections, and its center was marked in each section containing the label. We performed a similar analysis also for the double-labeled cells in V2 to determine their location relative to the center of the V2 injection site (Supplementary Fig. 3e). This data was plotted as histograms of the number of cells as a function of distance from the injection site center, as well as violin plots to illustrate the probability density distribution of spatial spreads.

**Reporting summary**. Further information on research design is available in the Nature Research Reporting Summary linked to this article.

## Data availability

The data that support the findings of this study are available from the corresponding author upon reasonable request. Source data are provided with this paper.

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

## Acknowledgements

We thank Kesi Sainsbury for technical assistance, Matthew Spurrier, Porter Babcock, Gabriella Rasmussen, and Alexander Ingold for help with imaging tissue sections. Supported by grants from the National Institute of Health (R01 EY026812, R01 EY019743, and BRAIN U01 NS099702), the National Science Foundation (IOS 1755431 and EAGER 1649923), and The University of Utah Neuroscience Initiative, to A.A., a grant from Research to Prevent Blindness, Inc. and a core grant from the National Institute of Health (EY014800) to the Department of Ophthalmology, University of Utah.

## Author contributions

Conceptualization: C.S., S.M., F.F., and A.A. Investigation: C.S., J.B., S.M., F.F., and A.A. Data analysis: C.S. and J.B. Writing-original draft: C.S. and A.A. Writing—review/editing: all authors. Visualization: C.S. Supervision and funding acquisition: A.A.

## Competing interests

The authors declare no competing interests.
