## [Peer Review File · Nature Communications]

Reviewers' comments:

Reviewer #2 (Remarks to the Author):

This study by Alessandra Angelucci's group, demonstrates directly that in macaque visual cortex feedback connections from V2 make monosynaptic contact with feedforward projecting neurons in V1 which are the source of their input to V2. The finding clarifies an element in the looped FF-FB circuit and contributes to the synaptic network underlying predictive coding. The findings presented in Figure 2 and 6 are solid and convincing. The results shown in Figures 3, 4, 5 and 7 rely on complex assumptions which makes their significance more difficult to evaluate.

Figures 1b-d are beautiful images which need additional explanation than what is available in text and legend. For example what is the relationship between (b) and (c)? After struggling with (c) it became clear that it is a zoomed-in image of Figure 2a which shows an injection site. Shouldn't there be two injection sites, one for AAV9-Flex-oG-mCherry/AAV9-Flex-TVA and the other for EnvA-RVGFP, unless they are perfectly matched? Figure 1b implies such a remarkable registration. The nearly invisible injection site in the CO image of Figure 2a supports this interpretation. If one maps the injection site of Figure 2a onto Figure 1c is surprising to see very few starter cells near the injection site and a fairly broad distribution of red cells indicating that CAV-cre exposed a large region of V2. If so, why then are starter cells clustered and the green cells relatively widely distributed? I suggest to revise Figure 1c to include the injections site(s) (see, Figure 2a). The legend could be improved by a more detailed description of the complex distributions of red, green and yellow labeled cells.

At the top of page 5 it is stated that GFP labeled cells within 355 um of the injection site are labeled due to "leak" of TVA-mCherry, but that generally mCherry is too weak to be detected in GFP labeled cells (Extended Figure 1). I understand that GFP labeled cells >400 um from the center depend on CAV-cre, but this does not exclude that many GFP labeled cells <400 um are monosynaptically connected to starter cells.

I do not understand how Figure 2g was generated. Figure 2a shows that the injection sites were ~1 mm apart. This implies that for MK405 and MK379 cells from each injection overlapped considerably. Could it be that many of the cells were double counted?

On page 5, para 4 it is stated that mCherry expression could only occur if V2 neurons were co-infected retrogradely with VAV2 and AAV9 vectors. This seems to contradict the argument about non-specific TVA-mCherry leak and goes against the interpretation that all double-labeled cells are specifically connected to CAV-cre injection sites within V2 and AAV9 injection sites in V1.

Figure 4e makes an important point showing that the spatial distribution of V2 FB projecting neurons is wider than the spread to V1->-V2 starter cells. How this was achieved needs to be explained in more detail. Convincing evidence needs to be presented which rules out that the reported spreads are not confounded by overlapping distributions of neuron labeled from neighboring injection sites. Further, it would be interesting to know how the spread translates into representation of visual space.

We are told that arrows in Figures 2a and 6a point to injection sites. With the exception of the upper left site in Figure 2a, these sites are difficult to confirm independently. I am aware how tricky a task this is, but the answer is important for interpreting the measurements shown in Figure 6f. The x-axis gives "Distance from the V1 injection center". Does that mean that the first column contains cells which are labeled by nonspecific leak of TVA-mCherry? The injections are so close together that one wonders how the distributions were kept apart and injection-specific distributions were extracted.

In Figure 6a it would be interesting to draw contours around blobs so that one could see the compartmental pattern. Having said that I realize that the pattern may be obscured by the multitude of injections.

Page 9, para 3, line 3. Please revise the sentence to indicate unambiguously that it remains unknown whether V1 FF neurons make monosynaptic contact with V2 FB neurons. - The point is made more clearly on page 10, para 2.

Page 9, last sentence. It may be premature to conclude stream-specificity without knowing which of the V2 compartments was injected with CAV2-cre. I suggest to de-emphasize the statement.

Page 10, para 1. The discussion give as succinct summary of the predictive coding framework. The central issue here is how prediction errors are computed. The key finding of the paper suggests that monosynaptic interactions are part of the underlying network architecture. If the network would be polysynaptic, what difference would that make?

An important point which is not addressed in the discussion is that V2 FB terminates in layers 1, 5 and 6 of V1. This highly likely involves monosynaptic contacts with apical dendrites of L2/3 and L5 pyramidal cells. The consequences of this organization for the computation of prediction error signals are substantial and need to be considered here to understand the significance of the present findings.

Page 10, para 4. I do not think that the data which suggest that the V2 FB originates from a 3.5 time larger region than the target in V1 are all that convincing. The main reservation comes from the challenge to assess the distribution of projection neurons from multiple, closely spaced injections.

Page 13, para 3. Please provide injection volumes.

Reviewer #3 (Remarks to the Author):

The authors have performed a heroic experiment, using an intersectional viral tracing strategy to map monosynaptic inputs to V1 neurons that project to V2 in macaque monkeys. Since this type of experiment is more difficult to do in monkeys than in rodents (where genetic Cre lines are available), it is a valuable result, particularly in contradistinction to similar experiments in the mouse. I believe the authors have met their "primary endpoint": they have shown an existence proof that some V2-projecting neurons in V1 ("feedforward" neurons) receive direct (i.e. monosynaptic) inputs from some V2 neurons that project back to V1 ("feedback" neurons). This is a worthwhile contribution. However, as will be detailed below, the authors often over-sell the results in a way that, if not corrected, will potentially cause more harm than good insofar as it leaves the reader with the wrong take-away message.

Major criticisms:

1. Overselling / Over-generalizing. This is rather pervasive throughout the manuscript, and I will not attempt to point out every instance. Suffice it to say that the authors need to tone down their claims given the limited nature of the experiment they did and the considerable methodological concerns. The main over-generalization is their claim that this represents a circuit motif that reflects FF/FB in general, when they have only tested a single instance, namely the interaction between V1 and V2, and only the central representation at that. This is not to detract from their main result, which I think is, on the whole, convincing. But V1 is an outlier in terms of cortical areas on nearly any dimension one considers, so it is risky to try to generalize this to all of cortex. Second, the "overselling" comes in statements such as (Abstract) ". . . we find that neurons sending FF projections to a higher-level area receive monosynaptic FB inputs exclusively from that area." (emphasis added). This is misleading in two important ways. First, it could be misconstrued as meaning that all of the inputs to the V2-projection neurons, when in fact these inputs represent, in the authors' own words, "only a tiny fraction" (p. 9) of the inputs to these neurons—most come from within V1. Second, even the authors intended meaning—i.e. that the monosynaptic FB inputs come from V2 but not from other extrastriate visual areas—is an oversell given the limitations of the technique and the intrinsic limits of interpreting a negative result. They say this based on the fact that in cases where they had V1 starter neurons, they found labeled neurons

in V2, but not in other extrastriate areas. But, to quote the old cliché, absence of evidence is not evidence of absence. And given the concerns about both false positives and false negatives, one needs to be more circumspect about how one interprets this finding. Another instance of overselling is the first sentence of the Discussion, which reads, "Using TRIO labeling in macaque visual cortex, we have shown that interareal FB connections to V1 selectively contact the V1 projection neurons that are the source of their interareal FF inputs." You cannot say that the V2-to-V1 FB neurons "selectively" contact V1-to-V2 projection neurons, because, by the design of the experiment, you don't know onto which other neurons these FB neurons might also make synapses. You'd need to do a completely different experiment to answer this question. I realize that this is not what the authors mean, but it would easily be misconstrued in the way I suggest. And, even though the next sentence helps to clarify the actual result, the authors still over-interpret the "lack of evidence" for such inputs from other extrastriate areas. As a final example, at the end of the 2nd paragraph of the Discussion, the authors state, "These findings strongly support the existence of area-, and thus, stream-specific, FF-FB loops in primate cortex." Again, there is the over-interpretation of the negative result, but, in addition, there is the problem of what the authors mean by "stream-specific." It is well established that there are different "streams" that interconnect V1 and V2 based on cytochrome oxidase compartments in the two areas, including one that ultimately connects to the dorsal stream (V1, layer 4B to V2 thick stripes to MT). Given that the authors don't know the location of their injections or label w/r/t cytochrome oxidase compartments, they need to be more circumspect here as well.

2. The Results section is very difficult to parse. I think much of this is because the authors don't clearly distinguish between caveats/controls and the main results. It might be clearer to have a separate section to address 1) the "non-specific infection of RVdG" due to the "leakiness" of TVA and 2) the problem of the "V2 Starter Cells." These could even be done after the main result (fig. 4) is presented, although I can see arguments for presenting the caveats before, as the authors have done. In either case, I think it would be helpful for the authors to say something more up front such as, "Before we get to our main findings, there are two limitations of our method that we need to address: 1 and 2. In the following two sections, we will specifically address these limitations and argue for why we think our results are still interpretable." Also, and this is a smaller point, it would be helpful to be more explicit in describing the "control" experiment. For example, at the bottom of p. 4 and top of p. 5, the authors state, "Control experiments (n=2) further demonstrated that all of the inputs arising from outside V1, and the vast majority of the intra-V1 inputs arising from beyond 400 μ m from the injection site were dependent on CAV2-Cre, . . ." It would be clearer to state exactly what the control was, as in "Control experiments (n=2), in which the CAV2-Cre injections were omitted, . . ."

3. In the summary diagram (fig. 8), the authors indicate (with a "?") that direct V2-to-V1 FB inputs to V1-to-V2 projection neurons from lower layer V2 neurons is uncertain. Yet, based on the reported results, I fail to see how this set of inputs is any more or less uncertain than that from superficial V2 neurons. What am I missing here?

4. Source of infragranular FB from V2 to V1. At the top of p. 7, the authors state, "Across the 3 cases, on average $54.3 \pm 10.2\%$ of GFP-labeled input cells were located in L5, $4.9 \pm 0.6\%$ in L6 and $40.1\% \pm 10.1\%$ in L2/3 (Fig. 4d), although the L5 bias in the population average is likely due to the intra-V2 inputs to the V2 starter cells in case MK379." Even with the potential contribution from "V2 starter cells," it concerns me that the vast majority of labeled FB cells in the infragranular layers of V2 were found in layer 5 and very few in layer 6. In all of the studies of which I'm aware, the vast majority of infragranular FB neurons are in layer 6. See, for example, figure 8 of Markov et al. 2014 (The Journal of Comparative Neurology 522:225–259), where there are a huge number of layer 6 FB neurons labeled after injection of a retrograde tracer in V1, and virtually none in layer 5. The authors argue that "the L5 bias in the population average is likely due to the intra-V2 inputs to the V2 starter cells in case MK379," but this alone does not seem to be able to account for such a large discrepancy. Can the authors make a more convincing quantitative argument here?

Minor criticisms:

1. Since the results vary across the 3 monkeys (particularly w/r/t the prevalence of V2 starter

cells), it would be helpful to include more detail about the monkeys (age, weight) and any other factors that might help account for differences, such as differences in the specific temporal intervals between the 3 injections and between the final injection and survival time.

2. Abstract. "In higher mammals, FF projections send afferents . . ." Grammar. "Projections" don't "send afferents." Should probably be "projection neurons send afferents."

3. Last sentence of 1st paragraph of Introduction. ". . . suggesting they play a fundamental computation, but their role remains poorly understood." Grammar. Either they "make a fundamental computation" or "play a fundamental computational role."

4. First sentence of 2nd paragraph of Introduction. "Traditional feedforward models of sensory processing postulate that FF connections mediate the complexification of RFs, and that object recognition occurs largely independently of FB signals, the latter purely serving strategic processing and attentional selection." What do the authors mean by "serving strategic processing"?

5. First paragraph of Results. "After about 3 weeks, necessary for the AAV genome to concatamerize, . . ." Most of the intended readership won't know what "concatamerize" means, so it should either be explained more fully or omitted.

6. middle of p. 5. ". . . starter cells were observed in all V1 layers known to send projections to V2 (layers 2/3, 4A, 4B, 5, 6), albeit the vast majority (~90%) of starter cells were located in L2/3." Grammar. "Albeit" should be "although."

7. middle of p. 7. "These results are consistent with previous reports that V2 FB neurons convey a larger region of visual space to their target V1 cells." Grammar. The neurons don't "convey" "a region of space." It would be more correct to say that they "convey information about a larger region of space." This same mistake is repeated at the bottom of p. 10.

-Rick Born

Reviewer #4 (Remarks to the Author):

Feedback (FB) and feedforward (FF) circuits are the fundamental characters of the mammalian cortex. Although many theories on cortical computation rely on inter-areal FF-FB connectivity, the anatomical substrates of FF-FB circuits at cellular level, in relation to FF neuron projection-dependent FB neuron connectivity, remain largely unknown. In this study, Siu et al attempt to solve this problem by applying a well-established circuit tracing method using modified rabies and AAVs to primate visual cortex. However, due to the quality (contamination) of samples and data, this study does not provide the answers to this important question. The results are often interpreted with assumptions, which may cause the bias and give little information to the field.

In the authors' experimental design, V1->V2 neurons are expected to serve as a sole starter neuron population, following CAV2-Cre injection in V2 and Cre-dependent AAV helper virus in V1. In all but one of their animals (MK382) this design did not meet this criterion. The authors record starter neurons within the V2 region, calling into question the source of monosynaptic spread for any given input neuron for the remaining experimental animals (MK379 & MK405). The authors claim that all or most input cells are presynaptic to the V1-> V2 starter neurons, saying 'tiny fraction' or 'safe to assume'. However, as there is no way to unmix the two distinct input populations (V1->V2 starter neurons, or 'leaky' V2 starter neurons), or to reliably predict the relative weight of input cells to each starter cells, these animal's data should be viewed as highly suspect. The remaining animal MK382 does provide useful data based on the designed experiment, however, n=1 data is little support to all the main claims of the manuscript.

Specific comments

1. The result section of this manuscript contains interpretation and assumptions without proper

evidence or support. These sentences have the potentials to bias the readers or to lead the conclusions not supported by the data. For example, on pg. 6, "therefore, it is safe to assume that all or most of the input cells, respectively, in the latter two cases are presynaptic to the V1->V2 starter neurons." In pg. 7, "As all V2 starter cells in this case were located in L5, it is likely that some fraction of the V2 L5 input cells were intra-laminar horizontal inputs presynaptic to the V2 starter cells, rather than FB inputs to the V1->V2 cells." On pg. 7, "L5 bias in the population average is likely due to the intra-V2 inputs to the V2 starter cells in case MK379." On pg. 7, "It is likely that the relatively larger number of GFP-labeled neurons in case MK379 was due to additional labeling of intrinsic V2 inputs to the presynaptic V2 starter cells found in this case. This interpretation is consistent with the evidence we present below that neurons receive the largest fraction of their inputs from cells located within the same cortical area." On pg. 8, "It is highly likely that these GFP-labeled cells in higher extrastriate cortex represent monosynaptic FB inputs to the V2->V1 starter cells. This indicates that a small fraction of FB connections, at least in higher cortical areas, can make direct contacts with FB-projecting neurons in lower-order areas, supporting the existence of cascading FB-to-FB projections connecting higher areas to V1 via a single synapse within each area." and more.

2. In discussion pg.9, "Our results demonstrate that FB connections selectively and monosynaptically contact neurons that are the source of their FF inputs. This is in contrast with results from mouse V1, where about 80-88% of FF projection neurons project to one or two higher visual areas, but only about 50% of their monosynaptic FB contacts arise from the same areas to which they project". The authors did not trace V2 FB axons to ensure that this statement is true. There could well be bifurcated axons reaching other areas from V2 FB neurons which are only identifiable by tracing rabies filled axons through serial reconstructions. This claim is weak in my view.

3. Optimization of virus volume, titer, and ratios are important for experimental success and reproducibility. For example, the successful animal MK382 received half as much virus as the two other attempts, these lower injection volumes should be considered for all future experiments to possibly reduce the chance of leaky infection to V2.

4. In Figure 3 and others, CAV2-cre is clearly causing necrotic damage in the V2 injection site. Damage in the target injection area may reduce the number of long-distance FB input neurons counted, especially considering retinotopically matched locations of FF and FB neurons in V1 and V2 respectively. Thus, another possibility of incorrect quantification.

5. In pg. 5, "Because in our control experiments 75% of all non-specific GFP label was located within 355 μm of the injection site center, in our quantitative analyses we omitted any GFP labeled cells within a 400 μm radius of the injection site." Is the exclusion of TVA leaky cells outside 400 μm sufficient? According to the author's observation, 25% of non-specific GFP cells are within from 400 μm to 1mm radius of the injection site. Importantly, AAV9-FLEX-TVA-mCherry + EnvA-RVdG-eGFP + CAV2-Cre but without AAV9-FLEX-oG should provide more proper controls to address this TVA leaky cell issues.

6. For extended data figure 1, which control animal is shown? High or low volume injections? Depending on that answer, the criteria to exclude TVA leaky GFP cells may need to be re-considered.

REPLY TO THE EDITOR AND REVIEWERS

We thank the Reviewers for their constructive criticism and the Editor for the opportunity to revise our manuscript. We have thoroughly revised the manuscript according to the Reviewers' comments, and hope they will find our revised version much improved. The main changes are as follows. In response to Reviewer 2 we have re-analyzed the spatial spread data in a manner that is independent of the location of the injection sites. In response to Reviewer 3, we have toned down the generality of our findings, expanded the description of the controls, discussed why TRIO is suited to estimate relative input strength, and provided additional evidence for the prevalence of layer 5 feedback to direct feedback-to-feedforward connections. In response to Reviewer 4, we provide additional evidence suggesting that in 1 of the 2 cases showing a few double-labeled cells in V2, the latter did not significantly contribute to trans-synaptic GFP label, and discuss more at length why our data is, thus, interpretable as is without needing additional experiments. To facilitate the review of the revised version, the main changes in the revised manuscript are highlighted in red. Below is a point-to-point reply to the Reviewers, with our answers in red text.

REVIEWER 2

1) This study by Alessandra Angelucci's group, demonstrates directly that in macaque visual cortex feedback connections from V2 make monosynaptic contact with feedforward projecting neurons in V1 which are the source of their input to V2. The finding clarifies an element in the looped FF-FB circuit and contributes to the synaptic network underlying predictive coding. The findings presented in Figure 2 and 6 are solid and convincing. The results shown in Figures 3, 4, 5 and 7 rely on complex assumptions which makes their significance more difficult to evaluate.

ANSWER 1. Most of the Reviewer's concerns seem to be focused on the distance measurements (former **Figs. 2g, 4e and 6f**). To address these concerns, we have proposed an alternative analysis (now in **Figs. 4c, 5c, 7f**). However, we would like to point out upfront that the distance measurement analysis could potentially be entirely removed from the paper and not detract from the main results/conclusions that V1→V2 cells receive direct monosynaptic inputs from V2 feedback (FB) and not from other extrastriate areas. The purpose of the distance measurements was primarily to obtain a measure of the convergence of FB projections, which, however, is not entirely novel information. It is well established that FB connections to V1 are highly divergent, in both primates (see for example Angelucci *et al.*, 2002) and mouse (Marques *et al.*, 2018). Our results in this study are consistent with previous results in primates from our own lab, and the only novel aspect of these results is that they describe the convergence of FB projections to a specific class of V1 projection neurons rather than to V1 cells in general, as in previous studies.

2) Figures 1b-d are beautiful images which need additional explanation than what is available in text and legend. For example what is the relationship between (b) and (c)? After struggling with (c) it became clear that it is a zoomed-in image of Figure 2a which shows an injection site.

ANSWER 2. The legend of former **Fig 1c** specified that this is an image of a V1 injection site through layers 2/3, whereas the legend of former **Fig. 2a** specified **Fig. 1c** was a higher magnification of the boxed region in **Fig 2a**. Unfortunately, due to a typo, the legend stated "*The region inside the white box is shown at higher magnification in panel (c)*", instead of saying in panel "*1c*". We apologize for this typo, and recognize that this created confusion.

Revision. To improve clarity, as suggested by the Reviewer, we have now reorganized **Figs. 1-3** as follows: **Fig. 1a,b** are now a single **Fig.1** which explains the method and experimental design. **Fig. 2** illustrates the V1 injection sites at low and high power (thus including panels c-d of former Fig. 1, and a-d of former Fig. 2). **Fig.3** illustrates the V2 injection sites (panels a-b of former Fig. 3 and a-b of former Fig. 4). Subsequent figures report the quantifications.

3) Shouldn't there be two injection sites, one for AAV9-Flex-oG-mCherry/AAV9-Flex-TVA and the other for EnvA-RVGFP, unless they are perfectly matched? Figure 1b implies such a remarkable registration. The nearly invisible injection site in the CO image of Figure 2a supports this interpretation. If one maps the injection site of Figure 2a onto Figure 1c is surprising to see very few starter cells near the injection site and a fairly broad

distribution of red cells indicating that CAV-cre exposed a large region of V2. If so, why then are starter cells clustered and the green cells relatively widely distributed? I suggest to revise Figure 1c to include the injections site(s) (see, Figure 2a). The legend could be improved by a more detailed description of the complex distributions of red, green and yellow labeled cells.

ANSWER 3. Former and new **Fig.1b** show the injection plan not the actual injection spread. We aimed to inject AAV9 and RV at the exact cortical locations and did so using blood vessels as reference, and Fig. 1b shows such a plan. In some instances, the injections indeed end up on top of each other or at least overlapped to such an extent that we could not distinguish them in CO staining as two separate “discolorations” (they fused); this is the case in the example injection sites indicated by the top-left arrow in **Fig. 2a top**. In other instances, however, the injections centers were more separated, so we could distinguish two nearby discolorations in CO staining, such as in the case of the middle and right injections (middle and right arrows) in **Fig. 2a top**; in the original figure we only pointed the arrow to one of the two V1 injections, but the Reviewer may notice that there are two additional small discolorations in CO immediately below the middle injection and immediately above the right injection. The locations of the V1 injection sites were further confirmed by the clustered GFP label, part of which is due to the local TVA leak. For ex, in **Fig.2a bottom**, one can clearly see three clusters of GFP label either coinciding with the location of the arrow (top left arrow) or just above the arrows (middle and bottom right arrows). In contrast, red label was only seen if the AAV injections were in retinotopic overlap with the CAV2 injection site in V2. In the case of **Fig. 2a**, only the middle injections were in good retinotopic correspondence with the CAV2 injection, which explains the fairly large cluster of red labeled cells in the middle of the field of view. Instead, only sparse red labeled cells are seen in correspondence of the left and right injection sites, indicating these AAV injections were not retinotopically well aligned to the V2 injection site; the sparse red cells near these injections are more likely labeled via retrograde AAV infection within V1 and retrograde CAV2 infection from V2. This is precisely why we made 3 injections in V1 rather than a single injection, i.e. to ensure that at least one of them matched retinotopically the V2 injection site.

The reason why there are much fewer starter cells than red and green cells is because for yellow cells to occur, the AAV and RV injection sites must overlap, and be in retinotopic correspondence with the V2 injection site. In the case of **Fig. 2a**, the middle AAV injection was retinotopically matched to the V2 injection site (therefore the dense red label), but the RV injection was displaced slightly anteriorly relative to the AAV injection, leading to only partial overlap of the two V1 injection sites; as a result, yellow starter cells only occurred in the region of overlap of the two V1 injections. In summary, given the injection plan, it is to be expected that red and green cells will outnumber the yellow starter cells.

Revision. We added two additional arrows to **Fig. 2a**, pointing to all visible AAV injection sites. Moreover, as suggested by the Reviewer, we added a detailed description of the distribution of red, green and yellow label in this case (see p. 5).

4) At the top of page 5 it is stated that GFP labeled cells within 355 μm of the injection site are labeled due to “leak” of TVA-mCherry, but that generally mCherry is too weak to be detected in GFP labeled cells (Extended Figure 1). I understand that GFP labeled cells >400 μm from the center depend on CAV-cre, but this does not exclude that many GFP labeled cells <400 μm are monosynaptically connected to starter cells.

ANSWER 4. Of course, GFP label at $<400\mu\text{m}$ of the injection site also results from trans-synaptic label from real V1 starter cells, but, as we cannot determine which GFP labeled cells are pre-synaptic to the starter cells versus caused by TVA “leak”, we omitted all GFP local label from our counts. This is exactly what should be done, as this method is not suited to map local connections (for a discussion of this point see: Callaway & Luo, 2015), but only long-range connections, and the purpose of our analysis here is to determine the long-range connections to V1 starter cells (see also **Answer 5 to Reviewer 4** below).

Revision. We have expanded the description of the TVA leak and related controls and make it clear that our TRIO protocol cannot be used to map local inputs but only long-range inputs (see p. 7-8 and p10 2nd paragraph).

5) I do not understand how Figure 2g was generated. Figure 2a shows that the injection sites were ~ 1 mm apart. This implies that for MK405 and MK379 cells from each injection overlapped considerably. Could it be that many of the cells were double counted?

ANSWER 5. Absolutely not, as each cell was only marked once and all we did was measure its distance from the nearest injection site. The injections are sufficiently separated that we can in fact detect them individually (again you can clearly see 3 GFP injection sites in **Fig. 2a bottom**, for example). We chose conservatively to measure the distance of each starter cell from the nearest injection site. However, the Reviewer is correct in thinking that, due to the proximity of the injection sites, we cannot be sure whether the more distant cells belong to one versus

the adjacent injection. Moreover, as some of the AAV infection occurs retrogradely, it is also possible that some of the more distant cells are located nearer to an adjacent injection site than to the injection site from which they arose, and therefore would be erroneously assigned to the nearer injection site. This implies that we could have somewhat underestimated the spread of the starter cell region.

Revision. To address this problem, we have performed a different analysis. We measured the full medio-lateral spread of the starter cell label in V1 (along an axis parallel to the V1/V2 border), irrespective of which injection site it arose from, and compared it to the full mediolateral spread of GFP label in V2. This analysis assumes that GFP label in V2 could result from any of the V2 starter cells in V1. We performed this analysis to estimate the spread of V1 and V2 double-labeled (yellow) cells, as well as V1 and V2 GFP label. The new results are reported in **Figs. 4c, 5c, 7f**, and described on pp. 6 3rd and 5th paragraphs, 7 1st paragraph, 9 1st paragraph, 10 3rd paragraph, 13 bottom paragraph to 14 1st paragraph, 19). This analysis did not alter our initial conclusions significantly, and is consistent with our previous results.

6) On page 5, para 4 it is stated that mCherry expression could only occur if V2 neurons were co-infected retrogradely with CAV2 and AAV9 vectors. This seems to contradict the argument about non-specific TVA-mCherry leak and goes against the interpretation that all double-labeled cells are specifically connected to CAV-cre injection sites within V2 and AAV9 injection sites in V1.

ANSWER 6. There is no contradiction. TVA leak does not lead to mCherry expression, but only to local GFP label [we only found 1 red cell in one control case, and 3 in the second case at the injected AAV site (**Supplementary Fig. 2**), and zero outside of the V1 injection site]. The reason for this is well known (Callaway & Luo, 2015; Schwarz *et al.*, 2015; Lavin *et al.*, 2020): mCherry expression due to leak is insufficient for mCherry to be detected; moreover, the levels of oG expression due to leak are far too low for trans-synaptic RVdG infection to occur. So all GFP label at the injected site due to leak consists of RVdG entering cells expressing small amount of TVA due to leak, which is only sufficient for RVdG to enter these cells and replicate in them producing GFP. This implies that mCherry expression in V2 can only occur if the cells contain Cre (which they can only express if they were infected retrogradely by the CAV2-Cre virus injected in V2), and are co-infected with AAV (which can infect these cells in V2 retrogradely, from their axon terminals in V1).

In fact, even some of the red cells in $V1 > 400 \mu\text{m}$ could result from retrograde infection by AAV, but this is not a confound for the purpose of our analysis and our conclusions, as such red cells would still be V1 neurons projecting to the V2 injection site, i.e. they are still $V1 \rightarrow V2$ projection neurons, just not infected at the soma.

Revision. We expanded the description of the controls and the concepts of TVA leak (see results pp. 7-8, and **Supplementary Fig. 2**) and of retrograde AAV infection leading to double-labeled cells in V2 (see results pp. 8 to p.10 1st paragraph, and **Supplementary Figs. 3-4**).

7) Figure 4e makes an important point showing that the spatial distribution of V2 FB projecting neurons is wider than the spread to $V1 \rightarrow V2$ starter cells. How this was achieved needs to be explained in more detail. Convincing evidence needs to be presented which rules out that the reported spreads are not confounded by overlapping distributions of neuron labeled from neighboring injection sites. Further, it would be interesting to know how the spread translates into representation of visual space.

ANSWER 7: See answers 1 and 5 above in which we describe how we have addressed the problem by performing a new analysis of spatial spread. With respect to translating these distances to visual field representation, our method does not allow us to do that. However, this was addressed in one of our previous studies (Angelucci *et al.*, 2002), which we now discuss in the Discussion section of this manuscript, pointing to how these cortical extents translate to visuotopic extents (p. 13 bottom paragraph to p.14 1st paragraph).

8) We are told that arrows in Figures 2a and 6a point to injection sites. With the exception of the upper left site in Figure 2a, these sites are difficult to confirm independently. I am aware how tricky a task this is, but the answer is important for interpreting the measurements shown in Figure 6f. The x-axis gives “Distance from the V1 injection center”. Does that mean that the first column contains cells which are labeled by nonspecific leak of TVA-mCherry? The injections are so close together that one wonders how the distributions were kept apart and injection-specific distributions were extracted.

ANSWER 8: While the “discolorations” in CO are indeed small, they appear consistently across sections, allowing us to determine injection site locations. In former **Fig. 6a** (now **Fig. 7a**) the arrows point at injection sites that we “tracked” down through the section stack from the superficial layers where they are better visible (we now state this in the figure legend). Additionally, the GFP leak at the injected site further confirms the location of these injection sites as defined in CO. So we used both criteria to define injection sites. However, the issue remains that

GFP-labeled V1 neurons at long-distances from the injection sites that produced them could be assigned to the wrong injection site if they happened to be closer to it.

Revision. As described in **answer 5** above, we have performed a different analysis by simply measuring the overall mediolateral spread of GFP label in V1 and comparing it to the spread of the V1 starter cells (see results p. 10 3rd paragraph, and **Fig. 7f**). In these measurements we excluded all neurons within 400µm from the injections sites defined as described above.

9) In Figure 6a it would be interesting to draw contours around blobs so that one could see the compartmental pattern. Having said that I realize that the pattern may be obscured by the multitude of injections.

ANSWER 9. In fact, the pattern is not obscured by the multitude of injections, because only one injection or two injections within the same stripe (spaced anteroposteriorly) were made in V2 and confined to a single stripe (we target injections to specific stripes using optical imaging). To be precise, the local V1 pattern of GFP label is obscured by the artifactual GFP label caused by TVA leak, but the long-range pattern is not obscured. However, the CO compartment specificity of these inputs is the subject of a separate paper we are working on, for which we need additional cases.

10) Page 9, para 3, line 3. Please revise the sentence to indicate unambiguously that it remains unknown whether V1 FF neurons make monosynaptic contact with V2 FB neurons. - The point is made more clearly on page 10, para 2.

ANSWER 10: In that sentence, we intended to say that it remains unknown whether FB connections make monosynaptic contacts with FF-projection neurons, not the opposite. Instead the paragraph on page 10 intends to say that it is unknown whether FF-projection neurons monosynaptically contact the cells from which they receive FB projections. So the two sentences convey two separate concepts and the latter concept to which the Reviewer is referring is extensively discussed in that second paragraph (p. 13 3rd paragraph).

11) Page 9, last sentence. It may be premature to conclude stream-specificity without knowing which of the V2 compartments was injected with CAV2-cre. I suggest to de-emphasize the statement.

ANSWER 11: This terminology has created confusion also for Reviewer 3, although by “stream-specificity” we did not imply CO-stripe specificity but area-specificity.

Revision. We removed the term “stream-specificity”, and use “area-specificity” instead.

12) Page 10, para 1. The discussion give as succinct summary of the predictive coding framework. The central issue here is how prediction errors are computed. The key finding of the paper suggests that monosynaptic interactions are part of the underlying network architecture. If the network would be polysynaptic, what difference would that make?

ANSWER 12: The predictive-coding framework does not strictly require monosynaptic FB-to-FF loops. Moreover, it requires at least one type of FB connections to exert suppression, implying contacts with inhibitory neurons. This is discussed in the same paragraph of the Discussion to which the Reviewer is referring (now p.12 bottom paragraph to p. 13 1st paragraph). However, the predictive-coding framework requires area-specificity of FF-FB and FB-FF interactions, i.e. the loops need to be area and stream specific. Thus, as V1 receives FB from multiple areas (and sends FF inputs to multiple areas), monosynaptic area-specific FB-to-FF contacts support the area-specificity of the FB-FF loops required by the theory. Moreover, monosynaptic FB to FF interactions imply that FB can directly (i.e. not requiring additional intra V1 processing) and more rapidly affect the responses of FF cells.

Revision. We have expanded the Discussion along the lines of this reply (pp. 13 1st paragraph)

13) An important point which is not addressed in the discussion is that V2 FB terminates in layers 1, 5 and 6 of V1. This highly likely involves monosynaptic contacts with apical dendrites of L2/3 and L5 pyramidal cells. The consequences of this organization for the computation of prediction error signals are substantial and need to be considered here to understand the significance of the present findings.

ANSWER 13: This is true. However, as we did not target viral injections to specific layers, we cannot parse out FB inputs to superficial vs deep layers.

Revision. We have added a discussion of the functional importance of FB terminations onto apical vs basal dendrites (p. 13 1st paragraph).

14) Page 10, para 4. I do not think that the data which suggest that the V2 FB originates from a 3.5 time larger

region than the target in V1 are all that convincing. The main reservation comes from the challenge to assess the distribution of projection neurons from multiple, closely spaced injections.

ANSWER 14: This is addressed in the answers above. Importantly, the conclusion has not changed significantly when applying the new analysis.

15) Page 13, para 3. Please provide injection volumes.

ANSWER 15: These were already reported in Supplementary Table 1, together with other injection parameters.

REVIEWER 3

The authors have performed a heroic experiment, using an intersectional viral tracing strategy to map monosynaptic inputs to V1 neurons that project to V2 in macaque monkeys. Since this type of experiment is more difficult to do in monkeys than in rodents (where genetic Cre lines are available), it is a valuable result, particularly in contradistinction to similar experiments in the mouse. I believe the authors have met their “primary endpoint”: they have shown an existence proof that some V2-projecting neurons in V1 (“feedforward” neurons) receive direct (i.e. mono-synaptic) inputs from some V2 neurons that project back to V1 (“feedback” neurons). This is a worthwhile contribution. However, as will be detailed below, the authors often over-sell the results in a way that, if not corrected, will potentially cause more harm than good insofar as it leaves the reader with the wrong take-away message.

Major criticisms:

1) Overselling / Over-generalizing. This is rather pervasive throughout the manuscript, and I will not attempt to point out every instance. Suffice it to say that the authors need to tone down their claims given the limited nature of the experiment they did and the considerable methodological concerns. The main over-generalization is their claim that this represents a circuit motif that reflects FF/FB in general, when they have only tested a single instance, namely the interaction between V1 and V2, and only the central representation at that. This is not to detract from their main result, which I think is, on the whole, convincing. But V1 is an outlier in terms of cortical areas on nearly any dimension one considers, so it is risky to try to generalize this to all of cortex.

ANSWER 1: Point taken, and we agree that, having only looked at the FB-to-FF loops between V2 and V1, we were overgeneralizing, albeit there was no intention of misleading.

Revision. We have revised text throughout the manuscript to specify that our findings are limited to the V1→V2 projection neurons and the FB connections to them (e.g. in the abstract, and discussion pp. 11, 12 2nd paragraph)

2) Second, the “overselling” comes in statements such as (Abstract) “. . . we find that neurons sending FF projections to a higher-level area receive monosynaptic FB inputs exclusively from that area.” (emphasis added). This is misleading in two important ways. First, it could be misconstrued as meaning that all of the inputs to the V2-projection neurons, when in fact these inputs represent, in the authors’ own words, “only a tiny fraction” (p. 9) of the inputs to these neurons—most come from within V1. Second, even the authors intended meaning—i.e. that the mono-synaptic FB inputs come from V2 but not from other extrastriate visual areas—is an oversell given the limitations of the technique and the intrinsic limits of interpreting a negative result.

ANSWER 2: The sentence in question, as grammatically constructed, is only consistent with the Reviewer’s 2nd interpretation, which is what we meant to say, but not with his first interpretation. We did not say “all inputs arise from V2”, rather we said “all monosynaptic FB inputs arise from V2”. The imprecision in the sentence is, again, perhaps in its generalization to FF and FB neurons, as opposed to V1 FF and V2 FB neurons.

Revision. That sentence now reads: “. . . we show that V1 neurons sending FF projections to area V2 receive monosynaptic FB inputs from V2, but not other V1-projecting areas.” For a discussion of interpreting negative results see point 3 below.

3) They say this based on the fact that in cases where they had V1 starter neurons, they found labeled neurons in V2, but not in other extrastriate areas. But, to quote the old cliché, absence of evidence is not evidence of absence. And given the concerns about both false positives and false negatives, one needs to be more circumspect about how one interprets this finding.

ANSWER 3: Here the Reviewer raises a more subtle issue. How is one supposed to interpret ANY anatomical study, if indeed absence of label cannot be interpreted as absence or sparsity of a projection? Anatomical studies have traditionally interpreted lack of label as absence of a projection or, allowing for the possibility of false negatives, as sparsity of a projection. In fact, recent evidence from the Surmeier's lab demonstrates that the TRIO method effectively allows to measure input strength. These authors found that the probability of RV trans-synaptic spread at each synapse is about 30% (Henrich *et al.*, 2020), thus, the amount of retrograde trans-synaptic label (in our study GFP) increases with the number of synapses formed by presynaptic neurons onto a given starter cell, providing an indirect measure of the functional strength of a projection (see Fig. 1 below- courtesy of Surmeier). Thus, we feel justified in interpreting absence of GFP label in other extrastriate areas as an indication that such projections are either absent or much sparser than the projection from V2, and so sparse that our labeling method fails to reveal them.

Revision. We are now more cautious in stating that there are no projections from other extrastriate areas and have added a discussion of the Surmeier's data illustrated in Fig. 1 below to the Discussion section (p. 12 2nd paragraph).

Figure 1. A model of RV spread. Probability that a cell in region X will be retrogradely labeled as a function of the number of synapses (N) it makes with a starter cell in region Y. P_{RV} : probability of RV spread. From Henrich *et al.* 2020 *Sci Adv.*

4) Another instance of overselling is the first sentence of the Discussion, which reads, “Using TRIO labeling in macaque visual cortex, we have shown that interareal FB connections to V1 selectively contact the V1 projection neurons that are the source of their interareal FF inputs.” You cannot say that the V2-to-V1 FB neurons “selectively” contact V1-to-V2 projection neurons, because, by the design of the experiment, you don’t know onto which other neurons these FB neurons might also make synapses. You’d need to do a completely different experiment to answer this question. I realize that this is not what the authors mean, but it would easily be misconstrued in the way I suggest. And, even though the next sentence helps to clarify the actual result, the authors still over-interpret the “lack of evidence” for such inputs from other extrastriate areas.

ANSWER 4: We agree, this sentence is not the most accurate way to describe our results, since the study was retrograde not anterograde.

Revision. Here and throughout the manuscript we have modified this statement as follows: “...we have shown that V1 neurons sending FF projections to V2 receive direct monosynaptic inputs from V2 FB neurons, but not from neurons in other extrastriate areas known to project to V1.” However, we also explain our rationale for why, lack of FB projections from higher extrastriate areas indirectly suggests that the FB only contact V1 neurons that are the source of their areal input (see p. 3 bottom paragraph to p. 4 1st paragraph). As far as how “lack of evidence” is interpreted, see Answer 3 above.

5) As a final example, at the end of the 2nd paragraph of the Discussion, the authors state, “These findings strongly support the existence of area-, and thus, stream-specific, FF-FB loops in primate cortex.”

Again, there is the over-interpretation of the negative result, but, in addition, there is the problem of what the authors mean by “stream-specific.” It is well established that there are different “streams” that interconnect V1 and V2 based on cytochrome oxidase compartments in the two areas, including one that ultimately connects to the dorsal stream (V1, layer 4B to V2 thick stripes to MT). Given that the authors don’t know the location of their injections or label w/r/t cytochrome oxidase compartments, they need to be more circumspect here as well.

ANSWER 5: The term “functional streams” is not only used with respect to CO stripes, but also to refer to dorsal vs. ventral stream, so areas are also part of streams. However, since this is obviously confusing for two of the Reviewers, we have removed any reference to “streams” and use “area-specificity or functional-specificity” instead.

6) The Results section is very difficult to parse. I think much of this is because the authors don’t clearly distinguish between caveats/controls and the main results. It might be clearer to have a separate section to address 1) the “non-specific infection of RVdG” due to the “leakiness” of TVA and 2) the problem of the “V2 Starter Cells.” These could even be done after the main result (fig. 4) is presented, although I can see arguments for presenting the caveats before, as the authors have done. In either case, I think it would be helpful for the authors to say something more up front such as, “Before we get to our main findings, there are two limitations of our method that we need to address: 1 and 2. In the following two sections, we will specifically address these limitations and argue for why we think our results are still interpretable.” Also, and this is a smaller point, it would be helpful to be more explicit in describing the “control” experiment. For example, at the bottom of p. 4 and top of p. 5, the authors state, “Control experiments (n=2) further demonstrated that all of the inputs arising from outside V1, and the vast majority of the intra-V1 inputs arising from beyond 400 μ m from the injection site were dependent on CAV2-Cre,” It would be clearer to state exactly what the control was, as in “Control experiments (n=2), in which the CAV2-Cre injections were omitted,”

ANSWER 6: We struggled with how to best organize the paper and indeed recognize that this needed to be reorganized in a way that improves flow.

Revision. We have reorganized the Results section as suggested by the Reviewer, i.e. by first presenting the main result (after initial description of the method) and then dealing with the issue of the TVA leak and the V2 starter cells under a section entitled “Control Experiments” (pp. 7-10). We have also expanded the discussion of the TVA leak and controls in the Results section (pp. 7-8).

7) In the summary diagram (fig. 8), the authors indicate (with a “?”) that direct V2-to-V1 FB inputs to V1-to-V2 projection neurons from lower layer V2 neurons is uncertain. Yet, based on the reported results, I fail to see how this set of inputs is any more or less uncertain than that from superficial V2 neurons. What am I missing here?

ANSWER 7: We apologize for the confusing figure. The two blue cells were not meant to indicate superficial vs deep layer FB cells, but just two different populations of cells in V2, one sending direct FB projections to V1→V2 neurons (that may be located in L2/3 or L5/6 or both), the other representing the starter cells in V2, which we only found in L5; the latter receive direct FB from extrastriate cortex, and may or may not send monosynaptic FB connections to V1→V2 cells.

Revision. We have modified Fig.8 (now Fig.9).

8) Source of infragranular FB from V2 to V1. At the top of p. 7, the authors state, “Across the 3 cases, on average $54.3 \pm 10.2\%$ of GFP-labeled input cells were located in L5, $4.9 \pm 0.6\%$ in L6 and $40.1\% \pm 10.1\%$ in L2/3 (Fig. 4d), although the L5 bias in the population average is likely due to the intra-V2 inputs to the V2 starter cells in case MK379.” Even with the potential contribution from “V2 starter cells,” it concerns me that the vast majority of labeled FB cells in the infragranular layers of V2 were found in layer 5 and very few in layer 6. In all of the studies of which I’m aware, the vast majority of infragranular FB neurons are in layer 6. See, for example, figure 8 of Markov et al. 2014 (The Journal of Comparative Neurology 522:225–259), where there are a huge number of layer 6 FB neurons labeled after injection of a retrograde tracer in V1, and virtually none in layer 5. The authors argue that “the L5 bias in the population average is likely due to the intra-V2 inputs to the V2 starter cells in case MK379,” but this alone does not seem to be able to account for such a large discrepancy. Can the authors make a more convincing quantitative argument here?

ANSWER 8: Our discussion of a L5 bias being due to the V2 starter cells was in relation to the relative proportion of FB inputs arising from L2/3 vs 5/6. Specifically, the 2 cases with no or few starter cells in V2 have an almost equal proportion of FB inputs from L2/3 and 5/6, while MK379 shows a relatively larger fraction of FB inputs from L5, which we think could be partly due to the contribution of inputs to the starter cells in V2 L5.

With respect to why the deep layer FB arises mostly from L5 and less so from L6, most previous reports show both L5 and 6 as sending FB projections from V2 to V1, but only few reports have quantified the relative proportion of L5 vs L6 FB. Indeed, as pointed out by the Reviewer, Markov et al (2014) show most cells as arising from L6, albeit note that the figures in that paper show plenty of cells in L5 at the border with L6. Rockland and Pandya (Rockland & Pandya, 1979; Rockland & Pandya, 1981) also show V2 FB as arising from both L5b and 6. It is certainly the case that assignment of exact layer borders is less accurate in tangential sections (as we have

used in our study) than in pia-to-white matter sections (as in the Markov study, for example), so we cannot exclude that a few cells at the border between L5 and L6 in our study were mis-assigned to one or the other layer. However, we are confident that the bulk of the V2 infragranular GFP label was located in L5.

Revision. We have added a new supplementary figure (**Suppl. Fig. 1**; cited on p. 6 of the Results) showing a series of sequential sections through V2 for one example case (MK405) encompassing the densest V2 infragranular GFP label. It is clear that the densest label is in L5. We have also added a discussion of the differences with previous results on the laminar distribution of FB neurons in the Discussion section (p. 12 3rd paragraph). In this discussion, we point out that the difference between our study and previous studies of V2 to V1 FB connections is that we have labeled FB projections to a specific V1 cell population, namely the cells that project to V2, whereas previous studies labeled all FB connections to V1. Therefore, it is possible that there is a laminar specialization in the deep layer FB projections, with those in L5 contacting preferentially V1→V2 neurons. However, should the Reviewer find our evidence of a L5 projection unconvincing, we could pool together L5 and 6 label into a single deep-layers label. But doing so may conceal an important layer specialization that was previously unknown.

9) Minor criticisms:

1. Since the results vary across the 3 monkeys (particularly w/r/t the prevalence of V2 starter cells), it would be helpful to include more detail about the monkeys (age, weight) and any other factors that might help account for differences, such as differences in the specific temporal intervals between the 3 injections and between the final injection and survival time.

ANSWER 9: We think that the factors that most affect the prevalence of starter cells in V2 and potential trans-synaptic RV infection from them are primarily the AAV volumes injected and the post-RV injection survival times, respectively. Larger volumes of AAV increase the probability of retrograde neuronal infection, explaining why we found starter cells in V2 in both MK379 and MK405 (both of which received larger volumes of AAV). Longer post-RV survival times, instead, may increase the probability of RV trans-synaptic infection and GFP expression in neurons sending inputs to the V2 starter cells. This could explain why MK379 (12d post-RV injection) showed some GFP label in extrastriate cortex and thalamus, while MK405 (10d post-RV) did not. The issue of the V2 starter cells is dealt in greater detail in **Answer 1 to Reviewer 4**.

Revision. We have added the weight of the animals to **Supplementary Table 1**, and the age of the animals to the Methods (p. 15), although we do not think those factors influence the presence of starter cells in V2. We have also added a discussion of what may influence the prevalence of starter cells in V2 to the Results (see p.8 bottom paragraph).

2. Abstract. “In higher mammals, FF projections send afferents . . . “ Grammar. “Projections” don’t “send afferents.” Should probably be “projection neurons send afferents.”

Revision. This was changed to “FF projection neurons”.

3. Last sentence of 1st paragraph of Introduction. “. . . suggesting they play a fundamental computation, but their role remains poorly understood.” Grammar. Either they “make a fundamental computation” or “play a fundamental computational role.”

Revision. This was changed to “they carry out a fundamental computation”.

4. First sentence of 2nd paragraph of Introduction. “Traditional feedforward models of sensory processing postulate that FF connections mediate the complexification of RFs, and that object recognition occurs largely independently of FB signals, the latter purely serving strategic processing and attentional selection.” What do the authors mean by “serving strategic processing”?

Revision. We removed “strategic processing”.

5. First paragraph of Results. “After about 3 weeks, necessary for the AAV genome to concatemerize, . . .” Most of the intended readership won’t know what “concatemerize” means, so it should either be explained more fully or omitted.

Revision. We now explain the meaning of the term (p.4). It essentially means that the virus makes many copies of its DNA sequence, which are bound together.

6. middle of p. 5. “. . . starter cells were observed in all V1 layers known to send projections to V2 (layers 2/3, 4A, 4B, 5, 6), albeit the vast majority (~90%) of starter cells were located in L2/3.” Grammar. “Albeit” should be “although.”

Revision. We now use “although”.

7. middle of p. 7. “These results are consistent with previous reports that V2 FB neurons convey a larger region of visual space to their target V1 cells.” Grammar. The neurons don’t “convey” “a region of space.” It would be more correct to say that they “convey information about a larger region of space.” This same mistake is repeated at the bottom of p. 10.

Revision. We have changed this sentence to “... that V2 FB neurons convey information from a larger region of visual space to their target V1 cells” (pp. 7, 14).

-Rick Born

Thank you Rick for reviewing our paper!

REVIEWER 4

Feedback (FB) and feedforward (FF) circuits are the fundamental characters of the mammalian cortex. Although many theories on cortical computation rely on inter-areal FF-FB connectivity, the anatomical substrates of FF-FB circuits at cellular level, in relation to FF neuron projection-dependent FB neuron connectivity, remain largely unknown. In this study, Siu et al attempt to solve this problem by applying a well-established circuit tracing method using modified rabies and AAVs to primate visual cortex.

ANSWER: The TRIO method is well established in the mouse, where Cre lines can be used. However, to our knowledge, this is the first report of the TRIO method applied to non-human primates (NHPs). Developing this method in NHPs was not straightforward. It required selecting the viral vectors with promoters that worked in this species (as several of the vectors proven to work in mouse did not work in the primate), as well as optimizing injection volumes and survival times.

1) However, due to the quality (contamination) of samples and data, this study does not provide the answers to this important question. The results are often interpreted with assumptions, which may cause the bias and give little information to the field.

In the authors’ experimental design, V1->V2 neurons are expected to serve as a sole starter neuron population, following CAV2-Cre injection in V2 and Cre-dependent AAV helper virus in V1. In all but one of their animals (MK382) this design did not meet this criterion. The authors record starter neurons within the V2 region, calling into question the source of monosynaptic spread for any given input neuron for the remaining experimental animals (MK379 & MK405). The authors claim that all or most input cells are presynaptic to the V1-> V2 starter neurons, saying ‘tiny fraction’ or ‘safe to assume’. However, as there is no way to unmix the two distinct input populations (V1->V2 starter neurons, or ‘leaky’ V2 starter neurons), or to reliably predict the relative weight of input cells to each starter cells, these animal’s data should be viewed as highly suspect. The remaining animal MK382 does provide useful data based on the designed experiment, however, n=1 data is little support to all the main claims of the manuscript.

Specific comments

1. The result section of this manuscript contains interpretation and assumptions without proper evidence or support. These sentences have the potentials to bias the readers or to lead the conclusions not supported by the data. For example, on pg. 6, “therefore, it is safe to assume that all or most of the input cells, respectively, in the latter two cases are presynaptic to the V1->V2 starter neurons.” In pg. 7, “As all V2 starter cells in this case were located in L5, it is likely that some fraction of the V2 L5 input cells were intra-laminar horizontal inputs presynaptic to the V2 starter cells, rather than FB inputs to the V1->V2 cells.” On pg. 7, “L5 bias in the population average is likely due to the intra-V2 inputs to the V2 starter cells in case MK379.” On pg. 7, “It is likely that the relatively larger number of GFP-labeled neurons in case MK379 was due to additional labeling of intrinsic V2 inputs to the presynaptic V2 starter cells found in this case. This interpretation is consistent with the evidence we present below that neurons receive the largest fraction of their inputs from cells located within the same cortical area.” On pg. 8, “It is highly likely that these GFP-labeled cells in higher extrastriate cortex represent monosynaptic FB inputs to the V2->V1 starter cells. This indicates that a

small fraction of FB connections, at least in higher cortical areas, can make direct contacts with FB-projecting neurons in lower-order areas, supporting the existence of cascading FB-to-FB projections connecting higher areas to V1 via a single synapse within each area.” and more.

ANSWER 1: We believe that we have sufficient evidence indicating that the contribution of V2 double-labeled cells to the GFP input label was significant only in one of the 3 cases, namely the case in which we used larger AAV injection volumes and longer post-RV injection times (MK379), and that when considered all together our data are interpretable. Our rationale and evidence is summarized below.

- 1) One of the 3 cases, MK382, which received the smallest AAV injection volume, had double-labeled cells only in V1. This case showed GFP-labeled cells in V2, thus, demonstrating unequivocally the existence of monosynaptic V2 FB contacts with V1→V2 cells.
- 2) In Case MK405, which received larger AAV volumes, but shorter post-RV injection survival times, only 2% of all double-labeled (yellow) cells (12 out of 575) were found in V2, in contrast to case MK379 in which 17% of double-labeled cells were found in V2. Importantly, for a double labeled cell to act as a real “starter cell”, it needs to be infected by all 4 vectors (CAV2, AAV-TVA-mCherry, AAV-oG, and RV); however, a cell can be double-labeled if infected only by 3 vectors (CAV2, AAV-TVA-mCherry, and RV), but not be a starter cell if it is not infected by AAV-oG, which is needed for trans-synaptic RV infection. Therefore, statistically only a fraction of the 12 double labeled cells in V2, in case MK405, are likely to be starter cells. In fact, several lines of evidence strongly suggest that the majority of these V2 double-labeled cells in this case were not starter cells. This evidence is presented below.
 - (i) Real starter cells are typically surrounded by GFP-labeled input cells. This is to be expected, because the majority of inputs to cortical neurons arise from their neighbors (Markov *et al.*, 2011). Accordingly, in all cases, starter cells in V1 were surrounded by clusters of GFP-labeled cells around them (e.g. see **Fig. 2b**). Such pattern of label was also observed for most V2 double-labeled cells in case MK379, but not for most V2 double-labeled cells in case MK405 (see **new Supplementary Fig. 4**).
 - (ii) Along the same line of reasoning, real starter cells are expected to receive a significant fraction of their inputs from neurons in the same cortical column in the layers that project to their home layer. In case MK405, most double-labeled cells in V2 were located in L2/3, which receives inputs from L4, therefore a substantial number of GFP label would be expected in L4. In contrast, we only found 3 GFP labeled cells in this layer (0.4% of total V2 GFP labeled cells; **Fig. 5a**).
 - (iii) The laminar distribution of GFP-labeled V2 FB neurons in case MK405 was virtually identical to that of case MK382, which had no starter cells in V2 (**Fig. 5a**). This was unlike case MK379, in which FB label shows a bias for L5, the layer where all V2 double-labeled cells were found in this case.
 - (iv) In case MK405, unlike case MK379, we found no GFP-labeled inputs outside of V1 and V2, indicating that no trans-synaptic RV infection of long-distance inputs to these V2 cells occurred.
- 3) Even in the case with a larger fraction (17%) of double-labeled neurons in V2, case MK379, we believe that the contribution of these neurons to the overall V2 GFP label was minor. This is because in this case only 7 GFP-labeled neurons were found in extrastriate cortex outside V2, suggesting a very limited infection of neurons presynaptic to these V2 double-labeled cells occurred. Based on our estimates in **Fig. 7g** showing that 91.6% of long-distance inputs arise from within the same cortical area, one can estimate that only 83 neurons potentially labeled from the V2 starter cells contributed to the overall GFP labeled found in V2 in case MK379. This amounts to only 7% of the total V2 label, suggesting 93% of GFP labeled cells in V2, even in this case, are presynaptic to the V1 starter cells.

Revision. We have greatly extended the discussion of the V2 starter cells in a new Results section entitled “*Retrograde AAV infection*” in which we discuss and document the evidence described above (pp. 8-10). We have also added a new figure, **Supplementary Fig. 4**, demonstrating the lack of local GFP label surrounding most of the V2 double-labeled cells in case MK405 as opposed to case MK379. While the Reviewer may think it straightforward to perform one additional experiment to replicate case MK382, in fact, we have already tried this in two additional animals, but failed because the smaller volumes used make it much more difficult to achieve overlap of the 4 vectors. We would continue to attempt to replicate this case, if we felt our results were not interpretable without an additional case. However, because of the arguments provided above, we believe our results are interpretable and, thus, an additional case, at the cost of many failures in a precious species, is not essential.

2) In discussion pg.9, “Our results demonstrate that FB connections selectively and monosynaptically contact neurons that are the source of their FF inputs. This is in contrast with results from mouse V1, where about 80-88% of FF projection neurons project to one or two higher visual areas, but only about 50% of their monosynaptic FB contacts arise from the same areas to which they project”. The authors did not trace V2 FB axons to ensure that this statement is true. There could well be bifurcated axons reaching other areas from V2 FB neurons which are only identifiable by tracing rabies filled axons through serial reconstructions. This claim is weak in my view.

ANSWER 2: We agree that the first sentence could be misleading (although we did not intend to mislead) as we performed a retrograde not an anterograde experiment, and therefore we have re-written it to imply that V1 neurons projecting to V2 receive monosynaptic FB inputs selectively from the same area to which they project, i.e. V2.

Revision. See Answer 4 to Reviewer 3.

3) Optimization of virus volume, titer, and ratios are important for experimental success and reproducibility. For example, the successful animal MK382 received half as much virus as the two other attempts, these lower injection volumes should be considered for all future experiments to possibly reduce the chance of leaky infection to V2.

ANSWER 3: MK379 was one of our first attempts. The protocol was modified in subsequent cases to optimize labeling and minimize unwanted label, such as TVA leak and V2 double-labeled cells. Smaller volumes of AAV injections, such as in MK382, while desirable for the lack of double-labeled cells in V2, make it considerably more difficult to achieve successful retinotopic overlap of all injections, particularly of the CAV2 and AAV injections, resulting in many failures. We have attempted to replicate this case and have failed in 2 out of 3 experiments. Case MK405 represents a compromise between too many failures and just a small number of V2 double-labeled cells most of which did not act as “real” starter cells, therefore not compromising interpretation of results.

4) In Figure 3 and others, CAV2-cre is clearly causing necrotic damage in the V2 injection site. Damage in the target injection area may reduce the number of long-distance FB input neurons counted, especially considering retinotopically matched locations of FF and FB neurons in V1 and V2 respectively. Thus, another possibility of incorrect quantification.

ANSWER 4: Based on our experience and that of others (e.g., Kremer E.J. and Callaway E., personal communications), CAV2 does not cause significant damage. The tiny damage visible in what is now **Supplementary Fig. 3a,b** is primarily mechanical damage caused by the injection pipette (which is typical of any anatomy experiment) and extends only about 100 μm . Virtually no damage is seen at the V2 injection site in **Fig.3a,b**, other than discolorations of CO, which is typically caused by the pipette, but not that lack of CO staining does not imply cell death, as CO is very sensitive to small traumas. Importantly, GFP-labeled cells in **Fig. 3** are seen in the immediate vicinity of the injection site, also indicating the extent of the damage caused by the pipette or the virus is minimal. The potential loss of GFP label at the center of the injected site is negligible, given the 6-13mm extent of the GFP label in V2 (**Fig. 5c**), and unavoidable (we use the smallest possible pipette tip diameter that does not cause clogging). Moreover, the point spread function in macaque V2 at parafoveal eccentricities is in the range of 1 mm or more, therefore potential loss of label within a 100-200 μm V2 region is truly negligible.

5) In pg. 5, “Because in our control experiments 75% of all non-specific GFP label was located within 355 μm of the injection site center, in our quantitative analyses we omitted any GFP labeled cells within a 400 μm radius of the injection site.” Is the exclusion of TVA leaky cells outside 400 μm sufficient? According to the author’s observation, 25% of non-specific GFP cells are within from 400 μm to 1mm radius of the injection site. Importantly, AAV9-FLEX-TVA-mCherry + EnvA-RVdG-eGFP + CAV2-Cre but without AAV9-FLEX-oG should provide more proper controls to address this TVA leaky cell issues. 6) For extended data figure 1, which control animal is shown? High or low volume injections? Depending on that answer, the criteria to exclude TVA leaky GFP cells may need to be re-considered.

ANSWER 5: Former Extended Data Fig. 1 showed averages from both control cases. We now show the spatial spread of the “leaky” GFP label separately for each of the two control cases in a new **Supplementary Fig. 2**. The case with the smaller injection volumes (MK381, panels a-c), in fact, showed a larger number of artifactual GFP label, likely because the latter depends not just on volume, but also on the amount of overlap between the AAV and RV injection sites (these are injected in the same location, but 3 weeks apart). In control case MK380 (panel d) 14% (n=16/117) of the “leaky” cells lay beyond 400 μm from the injection sites, while in case MK381 (panel

c) 20% of leaky cells are beyond 400 μ m (n=89/449). Given that the number of real GFP-labeled cells counted in V1 at distances >400 μ m amounted to 10,688 (MK405), 2,569 (MK382) and 7,112 (MK379), it would seem that the potential inclusion of 16-89 cells to these counts is negligible, amounting to max 0.8%, 3.5% and 1.3%, respectively of the total intra-V1 GFP label. Thus, including these cells in the counts does not alter any of our conclusions. In contrast, excluding 1 mm of V1 from these counts would exclude a significant fraction of real long-range horizontal inputs to the V1 starter cells. In macaque cortex, long-range connections are typically considered those beyond one functional hypercolumn, i.e. one full cycle of ocular dominance columns, ~800 μ m (we exclude 400 μ m on each side of the injection center, i.e. 800 μ m of V1).

We do not understand the purpose of the additional control suggested by the Reviewer. It seems to us that all that the suggested control would tell us is whether the long-distance GFP label in V1 and V2 is oG-dependent, rather than artifactual. However, we already know the answer to this question based on the control we performed; since we did not observe any long-distance GFP label when we omitted the Cre, we can be confident such label is the result of Cre-dependent oG expression. Thus, the additional control would seem to add little additional information and, as such, potentially a waste of a precious animal species. More importantly, understanding the details of the TVA leak to the point of possibly eliminating it, while an interesting endeavor for studies of the local circuitry, is beyond the scope of our study, whose goals were primarily to map the long-distance inter-areal inputs, and secondarily to compare these inputs to the long-distance intra-areal inputs.

Revision. We have modified former Extended Data Fig. 1 (now **Supplementary Fig. 2**) by showing results of each control case separately (see also results p. 7-8 for a detailed description of results from the two control cases). We have added a discussion of why excluding only the GFP label within 400 μ m of the injection is acceptable (p. 10 2nd paragraph).

REFERENCES CITED

- Angelucci, A, Levitt, JB, Walton, EJ, Hupe, JM, Bullier, J & Lund, JS. (2002). Circuits for local and global signal integration in primary visual cortex. *J Neurosci* **22**, 8633-8646.
- Callaway, EM & Luo, L. (2015). Monosynaptic Circuit Tracing with Glycoprotein-Deleted Rabies Viruses. *J Neurosci* **35**, 8979-8985.
- Henrich, MT, Geibl, FF, Lakshminarasimhan, H, Stegmann, A, Giasson, BI, Mao, X, Dawson, VL, Dawson, TM, Oertel, WH & Surmeier, DJ. (2020). Determinants of seeding and spreading of alpha-synuclein pathology in the brain. *Sci Adv* **6**.
- Lavin, TK, Jin, L, Lea, NE & Wickersham, IR. (2020). Monosynaptic Tracing Success Depends Critically on Helper Virus Concentrations. *Front Synaptic Neurosci* **12**, 6.
- Markov, NT, Misery, P, Falchier, A, Lamy, C, Vezoli, J, Quilodran, R, Gariel, MA, Giroud, P, Ercsey-Ravasz, M, Pilaz, LJ, Huissoud, C, Barone, P, Dehay, C, Toroczkai, Z, Van Essen, DC, Kennedy, H & Knoblauch, K. (2011). Weight consistency specifies regularities of macaque cortical networks. *Cereb Cortex* **21**, 1254-1272.
- Markov, NT, Vezoli, J, Chameau, P, Falchier, A, Quilodran, R, Huissoud, C, Lamy, C, Misery, P, Giroud, P, Ullman, S, Barone, P, Dehay, C, Knoblauch, K & Kennedy, H. (2014). Anatomy of hierarchy: feedforward and feedback pathways in macaque visual cortex. *J Comp Neurol* **522**, 225-259.
- Marques, T, Nguyen, J, Fioreze, G & Petreanu, L. (2018). The functional organization of cortical feedback inputs to primary visual cortex. *Nat Neurosci* **21**, 757-764.
- Rockland, KS & Pandya, DN. (1979). Laminar origins and terminations of cortical connections of the occipital lobe in the Rhesus monkey. *Brain Res* **179**, 3-20.
- Rockland, KS & Pandya, DN. (1981). Cortical connections of the occipital lobe in the rhesus monkey: interconnections between areas 17, 18, 19 and the superior temporal sulcus. *Brain Res* **212**, 249-270.
- Schwarz, LA, Miyamichi, K, Gao, XJ, Beier, KT, Weissbourd, B, DeLoach, KE, Ren, J, Ibanes, S, Malenka, RC, Kremer, EJ & Luo, L. (2015). Viral-genetic tracing of the input-output organization of a central noradrenaline circuit. *Nature* **524**, 88-92.

REVIEWERS' COMMENTS

Reviewer #2 (Remarks to the Author):

Congratulation to the authors for a thorough and successful revision of a strong manuscript. All of my concerns have been carefully considered and addressed by revisions of figures and text. I have no further comments.

Reviewer #3 (Remarks to the Author):

The authors have performed a heroic experiment, using an intersectional viral tracing strategy to map monosynaptic inputs to V1 neurons that project to V2 in macaque monkeys. Since this type of experiment is more difficult to do in monkeys than in rodents (where genetic Cre lines are available), it is a valuable result, particularly in contradistinction to similar experiments in the mouse. I believe the authors have met their "primary endpoint": they have shown an existence proof that some V2-projecting neurons in V1 ("feedforward" neurons) receive direct (i.e. monosynaptic) inputs from some V2 neurons that project back to V1 ("feedback" neurons). This is a worthwhile contribution and the authors have done a good job of revising the manuscript to address my concerns. I think this is now an excellent paper.

-Rick Born

Reviewer #4 (Remarks to the Author):

My original critiques were not well addressed in the authors responses. I added my comments to the each point the authors made below. In general, the strength of the conclusions that can be drawn from the limited data is weak. I hope my comments are helpful.

1. The result section of this manuscript contains interpretation and assumptions without proper evidence or support. These sentences have the potentials to bias the readers or to lead the conclusions not supported by the data. For example, on pg. 6, "therefore, it is safe to assume that all or most of the input cells, respectively, in the latter two cases are presynaptic to the V1->V2 starter neurons." In pg. 7, "As all V2 starter cells in this case were located in L5, it is likely that some fraction of the V2 L5 input cells were intra-laminar horizontal inputs presynaptic to the V2 starter cells, rather than FB inputs to the V1->V2 cells." On pg. 7, "L5 bias in the population average is likely due to the intra-V2 inputs to the V2 starter cells in case MK379." On pg. 7, "It is likely that the relatively larger number of GFP-labeled neurons in case MK379 was due to additional labeling of intrinsic V2 inputs to the presynaptic V2 starter cells found in this case. This interpretation is consistent with the evidence we present below that neurons receive the largest fraction of their inputs from cells located within the same cortical area." On pg. 8, "It is highly likely that these GFP-labeled cells in higher extrastriate cortex represent monosynaptic FB inputs to the V2->V1 starter cells. This indicates that a small fraction of FB connections, at least in higher cortical areas, can make direct contacts with FB-projecting neurons in lower-order areas, supporting the existence of cascading FB-to-FB projections connecting higher areas to V1 via a single synapse within each area." and more.

1) We believe that we have sufficient evidence indicating that the contribution of V2 double-labeled cells to the GFP input label was significant only in one of the 3 cases, namely the case in which we used larger AAV injection volumes and longer post-RV injection times (MK379), and that when considered all together our data are interpretable.

i) One of the 3 cases, MK382, which received the smallest AAV injection volume, had double-labeled cells only in V1. This case showed GFP-labeled cells in V2, thus, demonstrating unequivocally the existence of monosynaptic V2 FB contacts with V1->V2 cells.

Reviewer: This is correct. The authors do have a single working experiment which shows evidence of V2 FB connections with V1->V2 cells. A single example is hardly unequivocal evidence, at best most experiments of n=1 are considered preliminary results.

ii) In Case MK405, which received larger AAV volumes, but shorter post-RV injection survival times, only 2% of all double-labeled (yellow) cells (12 out of 575) were found in V2, in contrast to case MK379 in which 17% of double-labeled cells were found in V2. Importantly, for a double labeled cell to act as a real "starter cell", it needs to be infected by all 4 vectors (CAV2, AAV-TVA-mCherry, AAV-oG, and RV); however, a cell can be double-labeled if infected only by 3 vectors (CAV2, AAV-TVA-mCherry, and RV), but not be a starter cell if it is not infected by AAV-oG, which is needed for trans-synaptic RV infection. Therefore, statistically only a fraction of the 12 double labeled cells in V2, in case MK405, are likely to be starter cells. In fact, several lines of evidence strongly suggest that the majority of these V2 double-labeled cells in this case were not starter cells. This evidence is presented below.

(1) Real starter cells are typically surrounded by GFP-labeled input cells. This is to be expected, because the majority of inputs to cortical neurons arise from their neighbors (Markov et al., 2011). Accordingly, in all cases, starter cells in V1 were surrounded by clusters of GFP-labeled cells around them (e.g. see Fig. 2b). Such pattern of label was also observed for most V2 double-labeled cells in case MK379, but not for most V2 double-labeled cells in case MK405 (see new Supplementary Fig. 4).

Reviewer: The standard for determining a starter cell from a tracing experiment such as this is by colocalization of TVA and RV linked fluorophores. Any contest to that standard will by its nature require substantial evidence that colocalized cells are not in fact starters. Simply observing a lack of local inputs is an unsubstantiated inference. Histological detections of rabies glycoprotein should be provided as evidence.

(2) Along the same line of reasoning, real starter cells are expected to receive a significant fraction of their inputs from neurons in the same cortical column in the layers that project to their home layer. In case MK405, most double-labeled cells in V2 were located in L2/3, which receives inputs from L4, therefore a substantial number of GFP label would be expected in L4. In contrast, we only found 3 GFP labeled cells in this layer (0.4% of total V2 GFP labeled cells; Fig. 5a).

Reviewer: "Along the same line of reasoning" This is an extension of a series of assumptions without sufficient evidence to support the proposed hypothesis.

(3) The laminar distribution of GFP-labeled V2 FB neurons in case MK405 was virtually identical to that of case MK382, which had no starter cells in V2 (Fig. 5a). This was unlike case MK379, in which FB label shows a bias for L5, the layer where all V2 double-labeled cells were found in this case.

Reviewer: The starter cell leak is not present in MK382. The starter cell leak was most abundant in MK379. The starter cell leak was present but at a lower proportion in MK405. Here the authors state that in MK379 the input patterns were observably different than those in the successful experiment. In the same response the authors simply decide that while MK405 did indeed have starter cell leak, the contribution of those leaky starters is negligible or non-existent because the laminar distribution of inputs is similar. This is post hoc ergo propter hoc reasoning.

(4) In case MK405, unlike case MK379, we found no GFP-labeled inputs outside of V1 and V2, indicating that no trans-synaptic RV infection of long-distance inputs to these V2 cells occurred.

Reviewer: To this reviewer the result of MK405 is surprising. No inputs were found at all outside V1 or V2? Not only does this not prove the V2 starter cells were not the source of monosynaptic spread, this seems indicative of a failure of the rabies tracing system in this animal due to the inefficiency.

iii) Even in the case with a larger fraction (17%) of double-labeled neurons in V2, case MK379, we believe that the contribution of these neurons to the overall V2 GFP label was minor. This is because in this case only 7 GFP-labeled neurons were found in extrastriate cortex outside V2, suggesting a very limited infection of neurons presynaptic to these V2 double-labeled cells occurred. Based on our estimates in Fig. 7g showing that 91.6% of long-distance inputs arise from within the same cortical area, one can estimate that only 83 neurons potentially labeled from the

V2 starter cells contributed to the overall GFP labeled found in V2 in case MK379. This amounts to only 7% of the total V2 label, suggesting 93% of GFP labeled cells in V2, even in this case, are presynaptic to the V1 starter cells.

Reviewer: I would like to see cited sources to backup these claims, otherwise this argument is circular and based only on internal evidence and assumptions.

b) Revision. We have greatly extended the discussion of the V2 starter cells in a new Results section entitled "Retrograde AAV infection" in which we discuss and document the evidence described above (pp. 8-10). We have also added a new figure, Supplementary Fig. 4, demonstrating the lack of local GFP label surrounding most of the V2 double-labeled cells in case MK405 as opposed to case MK379. While the Reviewer may think it straightforward to perform one additional experiment to replicate case MK382, in fact, we have already tried this in two additional animals, but failed because the smaller volumes used make it much more difficult to achieve overlap of the 4 vectors. We would continue to attempt to replicate this case, if we felt our results were not interpretable without an additional case. However, because of the arguments provided above, we believe our results are interpretable and, thus, an additional case, at the cost of many failures in a precious species, is not essential.

Reviewer: This reviewer understands the difficulties of performing further experiments very well, likewise the difficulty of successfully implementing intersectional rabies tracing strategies. These methods often fail, and experiments must be repeated. For this reason, use of a precious species may not be advisable when lower mammals could provide similar insights. In the opinion of this reviewer the results are not interpretable as they stand beyond the data from MK382.

2) In discussion pg.9, "Our results demonstrate that FB connections selectively and monosynaptically contact neurons that are the source of their FF inputs. This is in contrast with results from mouse V1, where about 80-88% of FF projection neurons project to one or two higher visual areas, but only about 50% of their monosynaptic FB contacts arise from the same areas to which they project". The authors did not trace V2 FB axons to ensure that this statement is true. There could well be bifurcated axons reaching other areas from V2 FB neurons which are only identifiable by tracing rabies filled axons through serial reconstructions. This claim is weak in my view. ANSWER 2: We agree that the first sentence could be misleading (although we did not intend to mislead) as we performed a retrograde not an anterograde experiment, and therefore we have re-written it to imply that V1 neurons projecting to V2 receive monosynaptic FB inputs selectively from the same area to which they project, i.e. V2. Revision. See Answer 4 to Reviewer 3

3) Optimization of virus volume, titer, and ratios are important for experimental success and reproducibility. For example, the successful animal MK382 received half as much virus as the two other attempts, these lower injection volumes should be considered for all future experiments to possibly reduce the chance of leaky infection to V2.

a) ANSWER 3: MK379 was one of our first attempts. The protocol was modified in subsequent cases to optimize labeling and minimize unwanted label, such as TVA leak and V2 double-labeled cells. Smaller volumes of AAV injections, such as in MK382, while desirable for the lack of double-labeled cells in V2, make it considerably more difficult to achieve successful retinotopic overlap of all injections, particularly of the CAV2 and AAV injections, resulting in many failures. We have attempted to replicate this case and have failed in 2 out of 3 experiments. Case MK405 represents a compromise between too many failures and just a small number of V2 double-labeled cells most of which did not act as "real" starter cells, therefore not compromising interpretation of results.

Reviewer: This reviewer understands the difficulties of performing further experiments very well, likewise the difficulty of successfully implementing intersectional rabies tracing strategies. These methods often fail, and experiments must be repeated. For this reason, use of a precious species may not be advisable when lower mammals could provide similar insights. In the opinion of this reviewer the results are not interpretable as they stand beyond the data from MK382. Again, stronger evidence is required to show that standard starter cells are in fact not starter cells for monosynaptic rabies spread.

We are grateful for the opportunity to respond to the comments of Reviewer 4.

We would like to start by providing an overview of the issues being discussed with Reviewer 4, and why we think our data are important and interpretable. We have performed a “heroic” experiment in monkey (in the words of Reviewer 2) to address two main questions: 1. Do monosynaptic feedback (FB)-to-feedforward (FF) contacts exist in primate cortex? 2. And if so, are they area-specific, i.e. does FB selectively contact neurons sending FF projections to their home area? There is a consensus in the field of neuroscience that, while very difficult, these experiments are worth it, because it remains unclear to what extent what we learn from mouse studies applies to higher species and humans. We found that such FB-to-FF contacts exist, at least in V1, and that these contacts are much more area-specific than reported in mouse. Importantly, we found high area-specificity, despite limitations of our method which, in 2 out of 3 cases, produced, in addition to hundreds of double-labeled (DL, yellow) cells in V1, a handful of potentially confounding DL cells in V2. The potential confound is that some of the labeled (green) presynaptic input cells could represent inputs to the V2-DL cells instead of inputs to the V1-DL cells (whose inputs we wish to identify). In fact, we would argue that it is even more remarkable that we still find high area-specificity of FB-to-FF contacts, despite the presence of few V2-DL cells, because, if anything, the latter would be expected to label long-range inputs in extrastriate areas that project to V2, compromising area-specificity. Instead, we found labeled input cells only in V1 and V2 in 2 out of 3 cases, one of which had no V2-DL cells; moreover, even in the 3d case we found only 7 input cells in extrastriate cortical areas beyond V2. We argue that the reason we found few or no cells beyond V2 in the two cases with a handful of V2-DL cells is that most of these cells did not act as real starter cells, i.e. they did not lead to pre-synaptic GFP expression. Our main evidence in support of this interpretation is that most of these cells did not lead to any local GFP label; the latter **MUST** occur, if indeed the cells acted as starter cells, as it is well established that the vast majority of inputs to cortical cells arise from its immediate neighbors. Non-starter DL cells can occur if: (1) the cell is infected by only 3 of the 4 injected viruses (if AAV-oG does not infect the cell, the G-deleted RV cannot spread trans-synaptically), and/or (2) the survival time used was sufficient to lead to trans-synaptic spread of RV from soma-infected V1 neurons, but insufficient for trans-synaptic spread of RV from terminals-infected V2 neurons, as the latter result from retrograde AAV infection.

We acknowledge that our input cell counts are not 100% accurate, as they likely include some inputs to the V2-DL cells, but in our revised version of the manuscript we have provided solid arguments as to why the number of labeled inputs arising from the V2-DL cells is negligible.

Below is a point-to-point reply, with our answers in red.

Point 1. Reviewer: My original critiques were not well addressed in the authors responses. I added my comments to the each point the authors made below. In general, the strength of the conclusions that can be drawn from the limited data is weak. I hope my comments are helpful.

1. The result section of this manuscript contains interpretation and assumptions without proper evidence or support. These sentences have the potentials to bias the readers or to lead the conclusions not supported by the data. For example, on pg. 6, “therefore, it is safe to assume that all or most of the input cells, respectively, in the latter two cases are presynaptic to the V1->V2 starter neurons.” In pg. 7, “As all V2 starter cells in this case were located in L5, it is likely that some fraction of the V2 L5 input cells were intra-laminar horizontal inputs presynaptic to the V2 starter cells, rather than FB inputs to the V1->V2 cells.” On pg. 7, “L5 bias in the population average is likely due to the intra-V2 inputs to the V2 starter cells in case MK379.” On pg. 7, “It is likely that the relatively larger number of GFP-labeled neurons in case MK379 was due to additional labeling of intrinsic V2 inputs to the presynaptic V2 starter cells found in this case.

This interpretation is consistent with the evidence we present below that neurons receive the largest fraction of their inputs from cells located within the same cortical area.” On pg. 8, “It is highly likely that these GFP-labeled cells in higher extrastriate cortex represent monosynaptic FB inputs to the V2->V1 starter cells. This indicates that a small fraction of FB connections, at least in higher cortical areas, can

make direct contacts with FB-projecting neurons in lower-order areas, supporting the existence of cascading FB-to-FB projections connecting higher areas to V1 via a single synapse within each area.” and more.

1) We believe that we have sufficient evidence indicating that the contribution of V2 double-labeled cells to the GFP input label was significant only in one of the 3 cases, namely the case in which we used larger AAV injection volumes and longer post-RV injection times (MK379), and that when considered all together our data are interpretable.

i) One of the 3 cases, MK382, which received the smallest AAV injection volume, had double-labeled cells only in V1. This case showed GFP-labeled cells in V2, thus, demonstrating unequivocally the existence of monosynaptic V2 FB contacts with V1→V2 cells.

Reviewer: This is correct. The authors do have a single working experiment which shows evidence of V2 FB connections with V1→V2 cells. A single example is hardly unequivocal evidence, at best most experiments of n=1 are considered preliminary results.

Authors: The point is that this case (MK382) allowed us to establish the existence of area-specific, direct V2FB-to-V1FF contacts, facilitating interpretation of the two other cases. Based on this case, we know V1→V2 cells receive significant amounts of FB inputs from V2. Therefore, although in the other two cases some of the labeled inputs could have resulted from the V2-DL cells, we can be confident that, given the much larger number of V1-DL cells and the results of case MK382, not all labeled input cells in V2 resulted from the V2-DL cells. At most the error is in the input cell counts, not in the main finding. And this error is small (see below).

Point 2. ii) In Case MK405, which received larger AAV volumes, but shorter post-RV injection survival times, only 2% of all double-labeled (yellow) cells (12 out of 575) were found in V2, in contrast to case MK379 in which 17% of double-labeled cells were found in V2. Importantly, for a double labeled cell to act as a real “starter cell”, it needs to be infected by all 4 vectors (CAV2, AAV-TVA-mCherry, AAV-oG, and RV); however, a cell can be double-labeled if infected only by 3 vectors (CAV2, AAV-TVA-mCherry, and RV), but not be a starter cell if it is not infected by AAV-oG, which is needed for trans-synaptic RV infection. Therefore, statistically only a fraction of the 12 double labeled cells in V2, in case MK405, are likely to be starter cells. In fact, several lines of evidence strongly suggest that the majority of these V2 double-labeled cells in this case were not starter cells. This evidence is presented below.

(1) Real starter cells are typically surrounded by GFP-labeled input cells. This is to be expected, because the majority of inputs to cortical neurons arise from their neighbors (Markov et al., 2011). Accordingly, in all cases, starter cells in V1 were surrounded by clusters of GFP-labeled cells around them (e.g. see Fig. 2b). Such pattern of label was also observed for most V2 double-labeled cells in case MK379, but not for most V2 double-labeled cells in case MK405 (see new Supplementary Fig. 4).

Reviewer: The standard for determining a starter cell from a tracing experiment such as this is by colocalization of TVA and RV linked fluorophores. Any contest to that standard will by its nature require substantial evidence that colocalized cells are not in fact starters. Simply observing a lack of local inputs is an unsubstantiated inference. Histological detections of rabies glycoprotein should be provided as evidence.

(2) Along the same line of reasoning, real starter cells are expected to receive a significant fraction of their inputs from neurons in the same cortical column in the layers that project to their home layer. In case MK405, most double-labeled cells in V2 were located in L2/3, which receives inputs from L4, therefore a substantial number of GFP label would be expected in L4. In contrast, we only found 3 GFP labeled cells in this layer (0.4% of total V2 GFP labeled cells; Fig. 5a).

Reviewer: “Along the same line of reasoning” This is an extension of a series of assumptions without sufficient evidence to support the proposed hypothesis.

Authors: We have used the same TRIO approach used in previous studies in mouse by the Luo and Callaway laboratories, in which Cre is delivered via a retro vector (either CAV2 or AAVretro), and TVA and oG are delivered by two separate AAV vectors in different dilutions, with only the AAV9-TVA being linked to a fluorophore (Schwarz et al., 2015; Kim et al., 2020). As discussed on pp. 7-8 of the revised version of the manuscript, the advantage of this dual AAV approach is that it allows for reduced TVA expression (and thus reduced local TVA leak) but enhanced oG expression (needed for trans-synaptic RV infection). However, one limitation of this approach is that, due to the lack of a fluorophore linked to the oG, and the unavailability of an antibody against oG, oG cannot be identified in the DL cells; thus it is unknown whether all DL cells indeed are starter cells. In these previous studies this limitation is simply accepted, as is the fact that some of the AAV infection occurs retrogradely. In fact, these limitations are often not even acknowledged in these previous studies, and the few retrogradely infected DL cells often are not even reported. So, while the Reviewer is correct in stating that the standard has been to consider all DL cells as starter cells, this is inaccurate when TVA and oG are delivered by separate vectors. Unfortunately, we cannot identify the presence of oG histologically, because there isn't an antibody against oG.

However, most experiments are imperfect, and evidence based on established knowledge is often used to arrive at a correct interpretation of the data. We disagree with the Reviewer that “lack of local inputs is an unsubstantiated inference”. We also disagree with the Reviewer that there is not “sufficient evidence to support the proposed hypothesis”. It is well established in several species that the vast majority of inputs to cortical cells arise from local neighbors, located within a few hundred microns (Angelucci et al., 2002; Chisum et al., 2003; Binzegger et al., 2004; Markov et al., 2011). Therefore, a starter cell MUST label local input cells. Moreover, it is well known that in macaque visual cortex, and in particular V2, cells in layers (L)2/3 receive input from L4 (Valverde, 1978; Lund et al., 1981), therefore a real starter cell in V2 L2/3 MUST label inputs in L4. Lack of local and interlaminar input label, as we observed, therefore indicates that trans-synaptic infection of RV from those V2 DL cells did not occur. And because it has been demonstrated that the probability of RV trans-synaptic spread at each synapse is about 30% (Henrich et al., 2020), the number of labeled input cells increases with the number of synapses formed by presynaptic neurons onto a given starter cell, providing an indirect measure of the strength of a projection. As the strength of local projections to cortical neurons is much higher than that of long-distance projections (Markov et al., 2011), it is highly unlikely that long-distance inputs are labeled in the absence of local input label. This is consistent with the fact that we did not observe any input cells in extrastriate areas beyond V2 in 2 out of 3 cases, and found only 7 cells in the third case. This is not unsubstantiated inference without sufficient evidence. Rather, our arguments are founded on well-established connectivity data in monkey and other species, and strongly support the interpretation that all of the V2-DL cells in one case, and most of them in the second case, were not starter cells. There are at least two plausible reasons for why the V2-DL cells may not have been starter cells: 1. They were not infected with the AAV-oG, and/or 2. gene expression by retrograde infection takes longer than expression following anterograde infection, and the short survival times used allowed for the latter but not the former (see also introductory paragraph above). All of this is now discussed on pp. 8-9 of the revised manuscript version, and a new supplementary figure supporting our argument has been added (Supplementary Fig. 4).

Point 3: (3) The laminar distribution of GFP-labeled V2 FB neurons in case MK405 was virtually identical to that of case MK382, which had no starter cells in V2 (Fig. 5a). This was unlike case MK379, in which FB label shows a bias for L5, the layer where all V2 double-labeled cells were found in this case.

Reviewer: The starter cell leak is not present in MK382. The starter cell leak was most abundant in MK379. The starter cell leak was present but at a lower proportion in MK405. Here the authors state that in MK379 the input patterns were observably different than those in the successful experiment. In the same response the authors simply decide that while MK405 did indeed have starter cell leak, the contribution of those leaky starters is negligible or non-existent because the laminar distribution of inputs is similar. This is post hoc ergo propter hoc reasoning.

Authors: This is one of several arguments, not an isolated one, that we used in support of the interpretation that V2-DL cells in MK405 did not contribute significantly to the input label. Again our argument is based on established knowledge of connectivity. It is well established that intralaminar long-range horizontal connections are most prominent in layers 2/3 and 5. As the leaky V2-DL cells in MK379 were all located in L5, if they acted as starter cells, they should have produced long-range intralaminar inputs in V2 L5. And indeed, in this case we found a larger number of labeled input cells in V2-L5 compared to the other two cases, consistent with the interpretation that the L5V2 DL cells acted as starter cells in this case. In contrast, the similar distribution of V2 input label in the other two cases is one additional piece of evidence suggesting the few V2-DL cells in case MK405 did not act as starter cell.

Point 4: (4) In case MK405, unlike case MK379, we found no GFP-labeled inputs outside of V1 and V2, indicating that no trans-synaptic RV infection of long-distance inputs to these V2 cells occurred.

Reviewer: To this reviewer the result of MK405 is surprising. No inputs were found at all outside V1 or V2? Not only does this not prove the V2 starter cells were not the source of monosynaptic spread, this seems indicative of a failure of the rabies tracing system in this animal due to the inefficiency.

Authors: Here the Reviewer is contradicting his previous argument. Up to this point it seemed the Reviewer's concern was that the V2-DL cells may be the ONLY (or at least a significant) source of the input label, to the point that the results are uninterpretable. In V2 this amounts to 785 input cells (MK405) and 1,200 input cells (MK379). In V1, which is one synapse upstream of V2, this amounts to several thousand cells in both cases. However, the lack of label in areas one synapse downstream of V2 is now interpreted by the Reviewer as failure or inefficiency of the rabies tracing system. The large amounts of input cells found in V1 and V2, in fact, indicates the rabies tracing system worked well. Moreover, the laminar and area location of input cells in 2 of the cases is consistent with what would be expected based on the anatomy of V1 and V2. As discussed on p. 12 of the revised manuscript, the most parsimonious explanation is that the lack of extrastriate inputs in two cases indicates area-specificity of FB-to-FF contacts. In contrast, in the third case, the presence of extrastriate as well as thalamic inputs indicates these inputs resulted from the V2-DL cells, and their location is consistent with what we know about the connectivity of V2 (all these extrastriate areas are connected to V2, but only a subset is connected to V1). The fact that we only found few input cells beyond V2 in MK379 is also consistent with the low number of V2DL cells that would have given rise to these inputs.

Point 5: iii) Even in the case with a larger fraction (17%) of double-labeled neurons in V2, case MK379, we believe that the contribution of these neurons to the overall V2 GFP label was minor. This is because in this case only 7 GFP-labeled neurons were found in extrastriate cortex outside V2, suggesting a very limited infection of neurons presynaptic to these V2 double-labeled cells occurred. Based on our estimates in Fig. 7g showing that 91.6% of long-distance inputs arise from within the same cortical area, one can estimate that only 83 neurons potentially labeled from the V2 starter cells contributed to the overall GFP labeled found in V2 in case MK379. This amounts to only 7% of the total V2 label, suggesting 93% of GFP labeled cells in V2, even in this case, are presynaptic to the V1 starter cells.

Reviewer: I would like to see cited sources to backup these claims, otherwise this argument is circular and based only on internal evidence and assumptions.

Authors: These estimates were based on our own internal results demonstrating much more input cells in V1 than in V2. However, these estimates are entirely consistent with previously published studies which are discussed at point 2 above. In particular, the study by Markov et al. (2011) shows that about 80-85% of inputs to cortical cells arise from the same cortical area (see their Fig. 2A). Repeating these estimates using the Markov et al data leads to the same conclusion that the contribution of the few V2-DL cells in MK379 to the V2 input label was negligible.

Point 6: b) Revision. We have greatly extended the discussion of the V2 starter cells in a new Results section entitled “Retrograde AAV infection” in which we discuss and document the evidence described above (pp. 8-10). We have also added a new figure, Supplementary Fig. 4, demonstrating the lack of local GFP label surrounding most of the V2 double-labeled cells in case MK405 as opposed to case MK379. While the Reviewer may think it straightforward to perform one additional experiment to replicate case MK382, in fact, we have already tried this in two additional animals, but failed because the smaller volumes used make it much more difficult to achieve overlap of the 4 vectors. We would continue to attempt to replicate this case, if we felt our results were not interpretable without an additional case. However, because of the arguments provided above, we believe our results are interpretable and, thus, an additional case, at the cost of many failures in a precious species, is not essential.

Reviewer: This reviewer understands the difficulties of performing further experiments very well, likewise the difficulty of successfully implementing intersectional rabies tracing strategies. These methods often fail, and experiments must be repeated. For this reason, use of a precious species may not be advisable when lower mammals could provide similar insights. In the opinion of this reviewer the results are not interpretable as they stand beyond the data from MK382.

2) In discussion pg.9, “Our results demonstrate that FB connections selectively and monosynaptically contact neurons that are the source of their FF inputs. This is in contrast with results from mouse V1, where about 80-88% of FF projection neurons project to one or two higher visual areas, but only about 50% of their monosynaptic FB contacts arise from the same areas to which they project”. The authors did not trace V2 FB axons to ensure that this statement is true. There could well be bifurcated axons reaching other areas from V2 FB neurons which are only identifiable by tracing rabies filled axons through serial reconstructions. This claim is weak in my view. ANSWER 2: We agree that the first sentence could be misleading (although we did not intend to mislead) as we performed a retrograde not an anterograde experiment, and therefore we have re-written it to imply that V1 neurons projecting to V2 receive monosynaptic FB inputs selectively from the same area to which they project, i.e. V2. Revision. See Answer 4 to Reviewer 3

3) Optimization of virus volume, titer, and ratios are important for experimental success and reproducibility. For example, the successful animal MK382 received half as much virus as the two other attempts, these lower injection volumes should be considered for all future experiments to possibly reduce the chance of leaky infection to V2.

a) ANSWER 3: MK379 was one of our first attempts. The protocol was modified in subsequent cases to optimize labeling and minimize unwanted label, such as TVA leak and V2 double-labeled cells. Smaller volumes of AAV injections, such as in MK382, while desirable for the lack of double-labeled cells in V2, make it considerably more difficult to achieve successful retinotopic overlap of all injections, particularly of the CAV2 and AAV injections, resulting in many failures. We have attempted to replicate this case and have failed in 2 out of 3 experiments. Case MK405 represents a compromise between too many failures and just a small number of V2 double-labeled cells most of which did not act as “real” starter cells, therefore not compromising interpretation of results.

Reviewer: This reviewer understands the difficulties of performing further experiments very well, likewise the difficulty of successfully implementing intersectional rabies tracing strategies. These methods often fail, and experiments must be repeated. For this reason, use of a precious species may not be advisable when lower mammals could provide similar insights. In the opinion of this reviewer the results are not interpretable as they stand beyond the data from MK382. Again, stronger evidence is required to show that standard starter cells are in fact not starter cells for monosynaptic rabies spread.

Authors: We, and a large fraction of the neuroscience community, would disagree with the statement that experiments in primates are not advisable when one can get insights from lower mammals. There are countless examples in which the mouse brain, and even that of higher mammals such as cats or tree-shrews,

dramatically differ from that of monkeys, particularly in the visual system. It is precisely because of the need to confirm in monkeys results from lower species that efforts are being made in the community, including our laboratory, to develop viral tools that allow addressing circuit-specific and cell-type specific questions in primates. We strongly believe that experiments in primates are essential if we are to understand the human brain and develop treatments for human brain disorders. Reviewer 2 shares our beliefs and recognizes the importance of our study, precisely because it is the first attempt to apply circuit tracing methods to monkey cortex. Indeed, our results demonstrate much greater specificity of connections in monkey compared to mouse, emphasizing the need for primate studies.

REFERENCES

- Angelucci A, Levitt JB, Walton E, Hupé JM, Bullier J, Lund JS (2002) Circuits for local and global signal integration in primary visual cortex. *J Neurosci* 22:8633-8646.
- Binzegger T, Douglas RJ, Martin KA (2004) A quantitative map of the circuit of cat primary visual cortex. *J Neurosci* 24:8441-8453.
- Chisum HJ, Mooser F, Fitzpatrick D (2003) Emergent properties of layer 2/3 neurons reflect the collinear arrangement of horizontal connections in tree shrew visual cortex. *J Neurosci* 23:2947-2960.
- Henrich MT, Geibl FF, Lakshminarasimhan H, Stegmann A, Giasson BI, Mao X, Dawson VL, Dawson TM, Oertel WH, Surmeier DJ (2020) Determinants of seeding and spreading of alpha-synuclein pathology in the brain. *Sci Adv* 6:eabc2487.
- Kim EJ, Zhang Z, Huang L, Ito-Cole T, Jacobs MW, Juavinett AL, Senturk G, Hu M, Ku M, Ecker JR, Callaway EM (2020) Extraction of Distinct Neuronal Cell Types from within a Genetically Continuous Population. *Neuron* 107:1-9.
- Lund JS, Hendrickson AE, Ogren MP, Tobin EA (1981) Anatomical organization of primate visual cortex area VII. *J Comp Neurol* 202:19-45.
- Markov NT, Misery P, Falchier A, Lamy C, Vezoli J, Quilodran R, Gariel MA, Giroud P, Ercsey-Ravasz M, Pilaz LJ, Huissoud C, Barone P, Dehay C, Toroczkai Z, Van Essen DC, Kennedy H, Knoblauch K (2011) Weight consistency specifies regularities of macaque cortical networks. *Cereb Cortex* 21:1254-1272.
- Schwarz LA, Miyamichi K, Gao XJ, Beier KT, Weissbourd B, DeLoach KE, Ren J, Ibanes S, Malenka RC, Kremer EJ, Luo L (2015) Viral-genetic tracing of the input-output organization of a central noradrenergic circuit. *Nature* 524:88-92.
- Valverde F (1978) The organization of area 18 in the monkey. A Golgi study. *Anat Embryol (Berl)* 154:305-334.